# Neural coding of multiple motion speeds in visual cortical area MT

**Xin Huang\*, Bikalpa Ghimire, Anjani Sreeprada Chakrala, Steven Wiesner**

Department of Neuroscience, University of Wisconsin-Madison, Madison, United States

## eLife Assessment

This study concerns how macaque visual cortical area MT represents stimuli composed of more than one speed of motion. The study is **valuable** because little is known about how the visual pathway segments and preserves information about multiple stimuli, and the study involves perceptual reports from both humans and one monkey regarding whether there are one or two speeds in the stimulus. The study presents **compelling** evidence that (on average) MT neurons shift from faster-speed-takes-all at low speeds to representing the average of the two speeds at higher speeds. Ultimately, this study raises intriguing questions about how exactly the response patterns in visual cortical area MT might preserve information about each speed, since such information could potentially be lost in an average response as described here, depending on assumptions about how MT activity is evaluated by other visual areas.

**\*For correspondence:**
xin.huang@wisc.edu

**Competing interest:** The authors declare that no competing interests exist.

**Abstract** Motion speed is a salient cue for visual segmentation, yet how the visual system represents and differentiates multiple speeds remains unclear. Here, we investigated the encoding and decoding of multiple speeds. We first characterized the perceptual capacity of human and macaque subjects to segment overlapping stimuli moving at different speeds. We then determined how neurons in area MT of macaque monkeys represent multiple speeds. We found that the responses of MT neurons to two speeds showed a robust bias toward the faster speed component. This faster-speed bias occurred when both speeds were slow (≤20°/s) and diminished as stimulus speed increased. Our findings can be explained by a modified divisive normalization model, in which the weights for the speed components are proportional to the responses of a population of neurons (the weighting pool) with a broad range of speed preferences, elicited by the individual speeds. Regarding decoding, a classifier could distinguish MT responses to two speeds from those to a corresponding log-mean speed. We further found that it was possible to decode two speeds from the MT population response, supporting the theoretical framework of coding multiplicity in neuronal populations. The decoded speeds can account for perceptual performance in segmenting two speeds with a large (4x) but not a small (2x) separation. Our findings help define the neural coding rule of multiple speeds. The faster-speed bias in MT could benefit important behavioral tasks, such as figure-ground segregation, as figural objects tend to move faster than the background in the natural environment.

## Introduction

Neuroscientists have been investigating how neurons in the brain represent sensory information for decades. Previous studies have often focused on the neural coding of a single visual stimulus. However, natural environments are abundant with multiple entities that often co-occupy the receptive fields (RFs) of visual neurons. Segmenting visual objects from one another and their background is a

fundamental function of vision (*Braddick, 1993*). Yet, the neural mechanisms underlying the representation of multiple stimuli remain poorly understood. As the field moves toward understanding visual processing under more naturalistic conditions, it becomes increasingly important to uncover the principles by which the brain encodes multiple stimuli. Visual motion is a particularly salient cue for scene segmentation. Elements that share common motion are typically grouped into a single perceptual object, while entities moving at different velocities can often be segregated from each other. For instance, an object moving at a speed distinct from its background is more readily segmented. In this study, we investigated how the primate visual system represents multiple motion speeds.

The extrastriate area MT plays a crucial role in motion processing and motion-based segmentation (*Allman et al., 1985*; *Britten, 2003*; *Born and Bradley, 2005*; *Pasternak and Tadin, 2020*; *Born et al., 2000*; *Huang et al., 2007*; *Huang et al., 2008*). Segmentation of overlapping stimuli moving in different directions and speeds gives rise to the perception of transparent motion (*Braddick, 1997*; *Braddick et al., 2002*; *Mestre et al., 2001*; *Masson et al., 1999*). Previous studies have investigated how neurons in MT represent two directions of transparently moving stimuli (*Snowden et al., 1991*; *Qian et al., 1994*; *McDonald et al., 2014*; *Xiao et al., 2014*; *Xiao and Huang, 2015*; *Wiesner et al., 2020*; *Stoner and Albright, 1992*; *Krekelberg and van Wezel, 2013*). Although how cortical neurons represent the speed of a single stimulus has been well-studied (*Maunsell and Van Essen, 1983*; *Lisberger and Movshon, 1999*; *Nover et al., 2005*; *Pack et al., 2005*; *Krekelberg et al., 2006a*; *Perrone and Thiele, 2001*; *Priebe et al., 2003*; *Priebe et al., 2006*; *Liu and Newsome, 2003*), how neurons represent multiple speeds of transparently moving stimuli is largely unknown.

In characterizing how MT neurons represent multiple directions of transparently moving stimuli, we have previously shown that many neurons do not pool two directions equally, but weigh one direction more than the other (*Xiao and Huang, 2015*). We have also found that some MT neurons exhibit response nonlinearity when pooling two directions, in a manner that better represents the individual direction components. The heterogeneous response weights and response nonlinearity in representing multiple directions can benefit the neural coding of multiple stimuli (*Orhan and Ma, 2015*; *Xiao and Huang, 2015*), and may constitute an optimal population representation of visual motion with multiple directions (*Huang et al., 2017*). Unlike two motion directions for which the individual directions appear to be balanced in perceptual quality and salience, visual stimuli moving at two speeds appear to be asymmetrical – one slower and one faster. The goal of this study is to determine the neural coding principle for multiple speeds of overlapping stimuli.

Visual information is encoded in the brain by populations of neurons, and Bayesian inference provides a robust framework for understanding the population neural code (*Pouget and Snyder, 2000*; *Averbeck et al., 2006*; *Ma et al., 2006*; *Fiser et al., 2010*). Additionally, the visual system may be optimized to represent information in natural environments and to enhance performance in key behavioral tasks (*Barlow, 1961*; *Atick and Redlich, 1992*; *Simoncelli and Olshausen, 2001*; *Ganguli and Simoncelli, 2014*; *Manning et al., 2024*). Within this framework, we consider several scenarios for how MT neurons might encode two speeds within their RFs. (1) *Response averaging*: MT neurons may average the responses elicited by individual speed components, a phenomenon often observed in neural responses to multiple stimuli (e.g. *Recanzone et al., 1997*; *Zoccolan et al., 2005*). When the separation between the two speeds is smaller than the neuron's tuning width, the population response to two speeds would appear unimodal, peaking at an intermediate speed. While decoding two stimuli from a unimodal response is theoretically possible (*Zemel et al., 1998*; *Treue et al., 2000*), response averaging may result in poorer segmentation compared to encoding schemes that emphasize individual components, as demonstrated in neural coding of overlapping motion directions (*Xiao and Huang, 2015*). (2) *Bias toward the stronger component response*: A neuron may favor the speed component that elicits a stronger response, following a soft-max operation (*Riesenhuber and Poggio, 1999*). This scheme allows neurons to preferentially encode stimuli at speeds closer to their preferred speeds and maintain a population code that represents both components. (3) *Bias toward the slower speed component*: Given that slower speeds are more prevalent in natural environments (*Weiss et al., 2002*; *Stocker and Simoncelli, 2006*; *Zhang and Stocker, 2022*), MT neurons may favor slower components. Such encoding would align with the prior probability of natural speed distributions, optimizing for more frequent stimuli. (4) *Bias toward the faster speed component*: Neurons may prioritize faster-moving components. This scheme would enable better segmentation of a faster-moving stimulus from a slower background, facilitating the critical perceptual task of figure-ground

segregation. Finally, we investigated whether these encoding rules are dependent on stimulus speeds and the speed preferences of individual neurons.

Regarding neural decoding, previous studies successfully extracted single stimulus speeds from neuronal populations in MT using decoders, such as vector-averaging and maximum likelihood estimators (*Lisberger and Movshon, 1999*; *Cherian and Maunsell, 2025*; *Priebe and Lisberger, 2004*; *Huang and Lisberger, 2009*; *Yang and Lisberger, 2009*; *Krekelberg et al., 2006a*; *Krekelberg et al., 2006b*; *Krekelberg and van Wezel, 2013*). However, it is unclear whether simultaneously presented multiple speeds can be extracted from population neural responses, which would be difficult for decoders that only read out a single value. Zemel and colleagues developed a decoding framework that recovers the probabilistic distribution of a stimulus feature (*Zemel et al., 1998*; *Pouget et al., 2003*). Decoders of this type remain to be tested with neurophysiological and perceptual data.

We first characterized the perception of overlapping stimuli that moved simultaneously at two speeds. Our results showed that human and monkey subjects can segment overlapping stimuli based only on speed cues. The performance was better when the separation between the two stimulus speeds was larger, and the ability to segment speeds was reduced when stimulus speeds were fast. Next, we recorded neuronal responses from area MT of macaque monkeys. We made a novel finding that MT neurons exhibited a strong faster-speed bias when stimulus speeds were slow. As stimulus speeds increased, the faster-speed bias gradually shifted toward response averaging. We also demonstrated that a classifier could distinguish between a two-speed stimulus and a single-speed stimulus based on MT responses in a manner generally consistent with perception. We proposed a model in which each speed component was weighted by the responses of a population of neurons with a broad range of speed preferences elicited by that speed component. We also found that information about two speeds was carried in the population neural response in MT, and it was possible to extract either a single speed or two speeds in a way largely consistent with perception, with limitations when two stimulus speeds were less separated from each other. This study helps to fill a gap in understanding the neural coding principle of multiple motion speeds. It provides new insight into the mechanism underlying the neural representation of multiple visual stimuli.

## Results
### Perception of overlapping stimuli moving at different speeds
#### Human psychophysics

To establish the perceptual basis for our study, we first characterized how human subjects perceived overlapping stimuli moving at different speeds. We used similar visual stimuli in our psychophysics experiments as in our neurophysiology experiments. We asked how perceptual segmentation was impacted by the separation between two stimulus speeds and the mean stimulus speed.

The visual stimuli consisted of two overlapping random-dot patches presented within a stationary square aperture, 10° wide, and centered at an eccentricity of 11°. The random dots translated within the aperture in the same direction at two different speeds. It has been suggested that the neural encoding of speed in the visual cortex is on a logarithmic scale (*Maunsell and Van Essen, 1983*; *Lisberger and Movshon, 1999*; *Nover et al., 2005*). We used a fixed ratio between two speeds, which resulted in a fixed speed difference on the logarithmic scale. One set of stimuli had a 'large speed separation,' and the speed of the faster component was four times that of the slower component. The five speed pairs used were 1.25 and 5°/s, 2.5 and 10°/s, 5 and 20°/s, 10 and 40°/s, and 20 and 80°/s (*Figure 1B1*). Another set of stimuli had a 'small speed separation,' and the speed ratio was two. The five-speed pairs were 1.25 and 2.5°/s, 2.5 and 5°/s, 5 and 10°/s, 10 and 20°/s, and 20 and 40°/s (*Figure 1B2*). Experimental trials of bi-speed stimuli that had large and small speed separations were randomly interleaved.

Human subjects first performed a standard two-alternative forced-choice (2AFC) task to discriminate a bi-speed stimulus from the corresponding single-speed stimulus that moved at the log mean speed of the two component speeds. In each trial, the bi-speed and single-speed stimuli were presented in two consecutive time intervals in a random and balanced order (*Figure 1A*). At large (4x) speed separation, all four subjects could perform the task well when the component speeds were less than 20 and 80°/s (*Figure 1C1*). At 20 and 80°/s, the discrimination performance was poor (mean $d'$=0.74, standard error STE = 0.5), indicating that subjects could not segment the speed components.

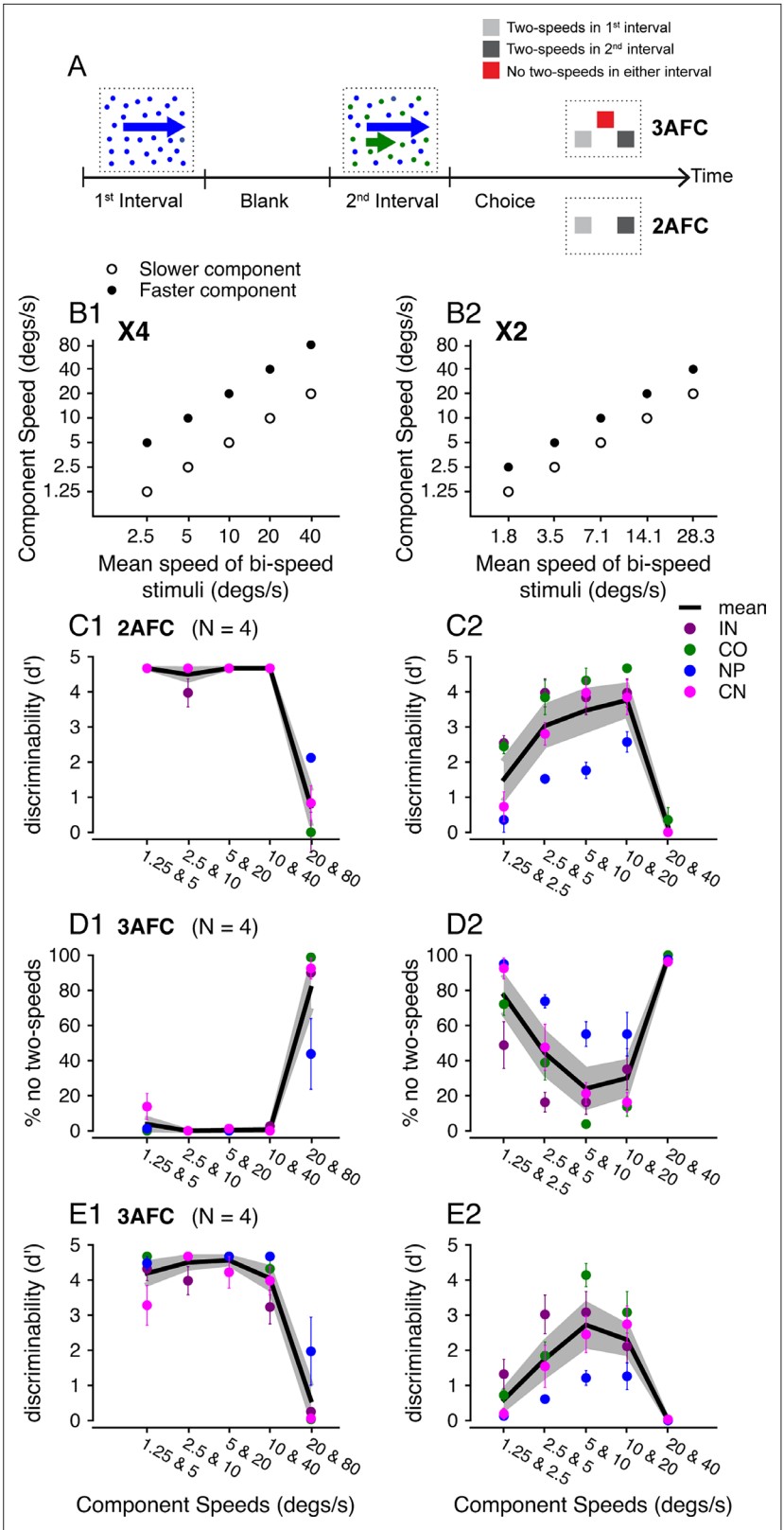

**Figure 1.** Psychophysical tasks and performance of human subjects. (**A**) Illustration of the 2AFC and 3AFC tasks. (**B**) Motion speeds of visual stimuli. The speeds of two stimulus components were plotted versus the log mean speed of each bi-speed stimulus. (**C**) Discriminability of four human subjects performing a standard 2AFC task. Letters are coded symbols for individual subjects. (**D**) In the 3AFC task, the percentage of trials that human subjects reported

*Figure 1 continued on next page*

*Figure 1 continued*

'no two-speeds.' (**E**) Discriminability of the same subjects performing the 3AFC task. (**B1–E1**) 4x speed separation. (**B2–E2**) 2x speed separation. Each color represents data from one subject. The solid line shows the subject-averaged result. Error bars and error bands represent ± STE. For each subject, five (n=5) blocks of experimental trials were tested, with 40 trials in each block (a total of 200 trials). Number of subjects (n = 4).

At the small (2x) speed separation, the discriminability was worse than at the 4x separation. When the component speeds were less than 20 and 40°/s, subjects on average could differentiate the bi-speed stimulus from the single-speed stimulus ($d'$>1.5), but not when speeds were at 20 and 40°/s (mean $d'$=0.17, STE = 0.1) (*Figure 1C2*).

In the standard 2AFC task, it is possible that subjects could not segment the bi-speed stimulus into two separate speeds but were still able to differentiate the bi-speed stimulus from the single-speed stimulus based on their appearance (e.g. the distribution of random dots in the bi-speed stimulus may appear less uniform). To address this concern, we designed a novel 3AFC task to measure discriminability based on perceptual segmentation. In the modified task, subjects still discriminated the bi-speed stimulus from the corresponding single-speed stimulus, but had the option to make a third choice on trials when they thought neither stimulus interval appeared to contain two speeds ('no two-speeds' choice) (*Figure 1A*). Panels D1 and D2 show the percentage of trials in which subjects made the no two-speeds choice (NTC). At 4x speed separation, the percentage of NTC was low at most speed pairs, except for the highest speeds of 20 and 80°/s (*Figure 1D1*). At 2x speed separation, the percentage of NTC showed a U-shape as a function of the stimulus speed and was near 100% at 20 and 40°/s (*Figure 1D2*). These results confirmed that human subjects had difficulty segmenting two speeds when stimulus speeds were high. In addition, at low stimulus speeds with a small (2x) speed separation, subjects tended to perceive only one speed (*Figure 1D2*). We incorporated the NTC into the $d'$ calculation by evenly splitting the NTC trials into 'hit' trials and 'false alarm' trials (see Methods). In this way, the NTC trials were accounted for by $d'$, in the sense that they did not contribute to successful discrimination.

The $d'$ from the 3AFC task were similar to those of the 2AFC task, with a slight reduction across conditions as the NTC trials reduced discrimination performance (*Figure 1E1 vs 1C1, 1E2 vs 1C2*). The small performance difference between the 2AFC and 3AFC tasks suggests that human subjects generally relied on speed segmentation to perform the 2AFC task. Based on the results from the 3AFC task, we performed a two-way ANOVA, with the two factors being the mean speed of the stimulus components and the speed separation (4x or 2x). We found that both factors had significant effects. $d'$ changed significantly with the mean stimulus speed (F(4,30) = 26.8, $p$=1.60 × 10$^{-9}$) and the $d'$ at 4x separation differed significantly from that at 2x separation (F(1,30) = 84.1, $p$=3.29 × 10$^{-10}$). $d'$ was higher at 4x than at 2x speed separation except at the fastest speeds of 20 and 80°/s vs. 20 and 40°/s (*Figure 1E1 vs 1E2*). Our results also showed that segmentation was significantly worse at fast speeds – $d'$ dropped significantly as the stimulus speeds increased from 10 and 40°/s to 20 and 80°/s for 4x separation (one-way ANOVA, F(1,6) = 38.6, $p$=8.1 × 10$^{-4}$) (*Figure 1E1*), and from 10 and 20°/s to 20 and 40°/s for 2x separation (one-way ANOVA, F(1,6) = 32.7, $p$=1.24 × 10$^{-3}$) (*Figure 1E2*).

## Monkey psychophysics

We next measured the monkey's ability to segment overlapping stimuli moving at two speeds. We trained one male macaque monkey to perform a 2AFC task, in which it reported whether a stimulus contained one or two speeds (*Figure 2A*, see Methods). The monkey's performance at 2x speed separation (*Figure 2B2*) was very similar in shape to that of humans (*Figure 1C2* of the 2AFC task). In addition, the monkey's performance was generally better at 4x separation than at 2x separation (*Figure 2B1 vs B2*).

At 4x separation, performance improved as stimulus speeds increased from 1.25 and 5°/s to 5 and 20°/s. As the stimulus speeds increased from 5 and 20°/s to 20 and 80°/s, the performance declined (*Figure 2B1*), similar to the human results (*Figure 1C1*). However, the monkey was still able to differentiate the bi-speed and single-speed stimuli at the fastest speeds of 20 and 80°/s (*Figure 2B1*), whereas the average human performance was poor (*Figure 1C1*). Note that one human subject (NP) performed better than other subjects at 20 and 80°/s (mean $d'$=2.12, STE = 0.12) (*Figure 1C1*). The difference between the monkey and human results may be due to species differences or individual variability. The

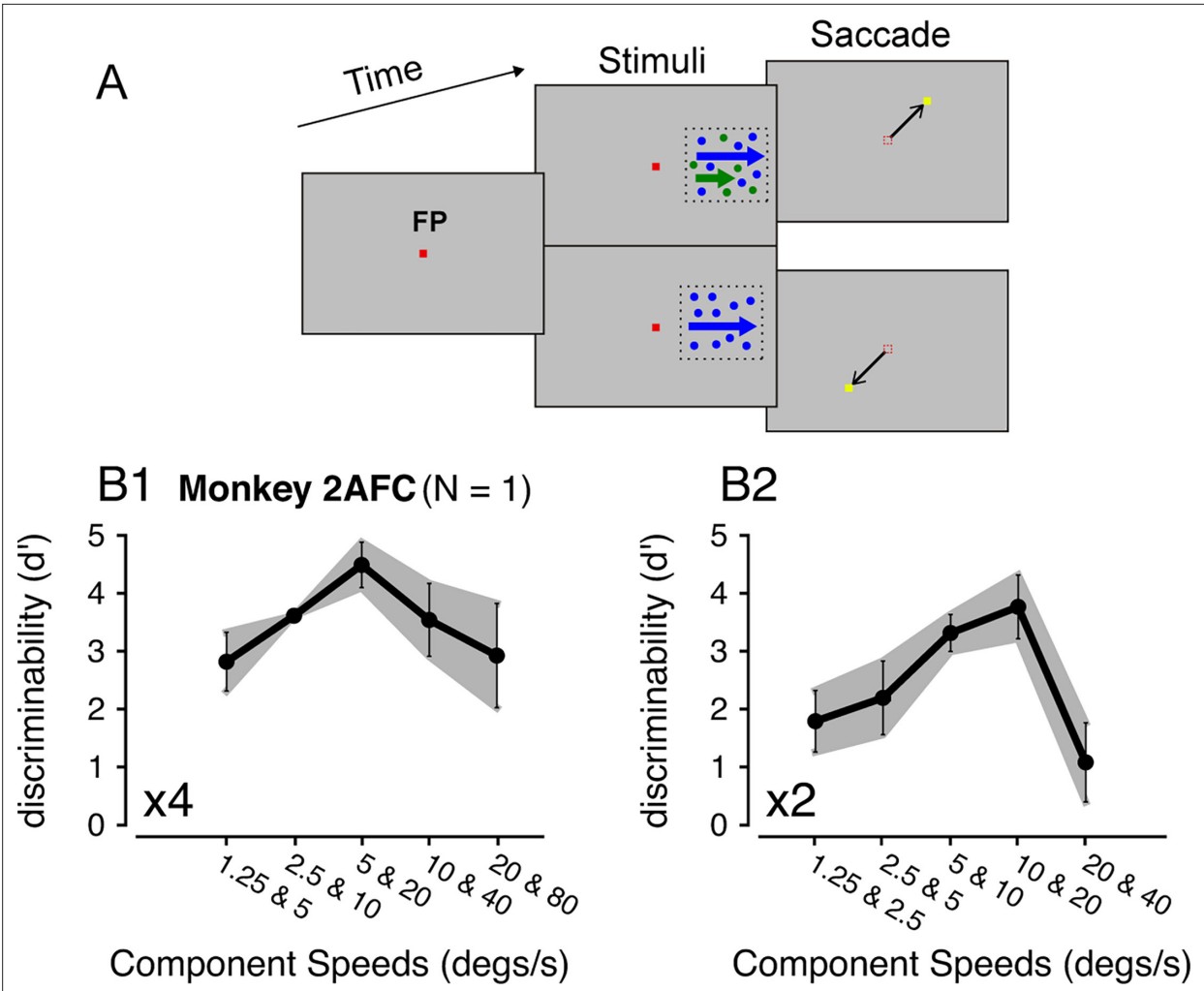

**Figure 2.** Monkey psychophysics. (**A**) The behavioral task and visual stimuli. (**B**) Discriminability of a monkey subject performing a 2AFC task. (**B1**) 4x speed separation. Total 50 trials in 5 sessions for each speed pair. (**B2**) 2x speed separation. Total 90 trials in 9 sessions for each speed pair. Error bars and error bands represent ± STE (n = 5 for 4x, n = 9 for 2x speed separation).

differences in behavioral tasks may also play a role – the monkey received feedback on the correctness of the choice (except for 20 and 80°/s and 20 and 40°/s), whereas human subjects did not.

Another notable difference between the monkey and human results was that, at low stimulus speeds of 1.25 and 5°/s, human subjects could differentiate the bi-speed stimulus from the corresponding single-speed (2.5°/s) stimulus nearly perfectly. In comparison, the ability of the monkey subject to segment 1.25 and 5°/s was lower ($d'$=2.8, STE = 0.51), although still good (*Figure 2B1* vs *Figure 1C1*). This may be explained by how the monkey performed the task. For human subjects, while the motion of the faster component (5°/s) of the bi-speed stimulus appeared to be salient, it required effort to notice the very slow component (1.25°/s) to be moving rather than stationary. In some trials, the monkey might be able to segment the 5°/s component from the bi-speed stimulus but consider the slower component of 1.25°/s as stationary and, therefore, reported that the stimulus contained only one speed. Despite some differences between the human and monkey results, the two general trends – better segmentation performance at larger than smaller speed separation and reduced segmentation ability at very fast speeds were consistent across species.

## Neuronal responses in MT elicited by bi-speed stimuli and single-speed components

To characterize how neurons in the visual cortex encode overlapping stimuli moving at different speeds, we recorded extracellularly from 100 isolated neurons in area MT of two male macaque monkeys (60 neurons from IM and 40 neurons from MO) while they performed a fixation task. *Figure 3* shows the responses from four example neurons. To visualize the relationship between the responses to the bi-speed stimulus (red) and the constituent speed components, the plots of the response tuning curves to the slower (green) and faster (blue) components are shifted horizontally so that the responses elicited by the bi-speed stimulus and its constituent single-speed components are aligned along a vertical line, as illustrated in *Figure 3A1*.

We found that the relationship between the responses elicited by the bi-speed stimulus and the constituent components depended on the stimulus speeds. *Figure 3A1–D1* shows the neuronal responses when the speed separation was large (4x). The component speeds were the same as the bi-speed stimuli used in the psychophysics experiments. When the two component speeds were slow (1.25 and 5°/s), the response to the bi-speed stimulus nearly followed the response elicited by the faster-speed component (the leftmost data points in *Figure 3A1–D1*). Importantly, the response elicited by the bi-speed stimuli did not simply follow the stronger component response. When the preferred speed of a neuron was sufficiently low such that the response elicited by the faster component was weaker than that elicited by the slower component, the response to the bi-speed stimulus still followed the weaker response elicited by the faster component (*Figure 3A1*). When the speeds of the two stimulus components were at 2.5 and 10°/s, the response elicited by the bi-speed stimulus was also biased toward the faster component, albeit to a lesser degree. As the mean speed of the two stimulus components increased, the bi-speed response became closer to the average of the two component responses (*Figure 3A1–D1*). We found similar results when the speed separation between the two components was small (2x) (*Figure 3A2–D2*).

We found the same trend in the neural responses averaged across 100 neurons (*Figure 4A*). At 4x speed separation, the population-averaged response showed a strong bias toward the faster component when the stimulus speeds were low and shifted toward the average of the component responses as the speeds increased (*Figure 4A1*). To determine whether this trend held for neurons with different preferred speeds, we divided the neuron population into three groups with 'low' (<2.5°/s), 'intermediate' (between 2.5 and 25°/s), and 'high' (>25°/s) preferred speeds. For 10 neurons that preferred low speeds, the response to the faster component was weaker than that to the slower component. However, the response to the bi-speed stimuli was strongly biased toward the faster component when the stimulus speeds were low (*Figure 4B1*). This finding suggests that the bi-speed response is not biased toward the stimulus component that the neuron prefers when presented alone, but biased toward the faster speed component.

For 61 neurons that preferred intermediate speeds (*Figure 4C1*) and 29 neurons that preferred high speeds (*Figure 4D1*), we also found a strong bias toward the faster speed component when the stimulus speeds were low, and a gradual change toward the average of the component responses as the stimulus speeds increased. At the lowest stimulus speeds of 1.25 and 5°/s, the bi-speed response was nearly identical to that elicited by the faster component, showing 'faster-component-take-all.' For neurons that preferred high speeds, faster-component-take-all was also found for the stimulus speeds of 2.5 and 10°/s (*Figure 4D1*). At the fastest speeds of 20 and 80°/s, the response to the bi-speed stimuli showed a slight bias toward the slower component (*Figure 4D1*). We found similar results at 2x speed separation (*Figure 4A2–D2*), although the effect is not as pronounced as 4x speed separation.

## Relationship between the responses to bi-speed stimuli and constituent stimulus components

We aimed to quantify the relationship between the response elicited by the bi-speed stimuli and the corresponding component responses. We first assumed that the response $R$ of a neuron elicited by two component speeds can be described as a weighted sum of the component responses $R_s$ and $R_f$ elicited by the slower ($v_s$) and faster ($v_f$) component speed, respectively *Equation 1*.

$$R\left(v_s, v_f\right) = w_s\left(v_s, v_f\right) \bullet R_s + w_f\left(v_s, v_f\right) \bullet R_f, \tag{1}$$

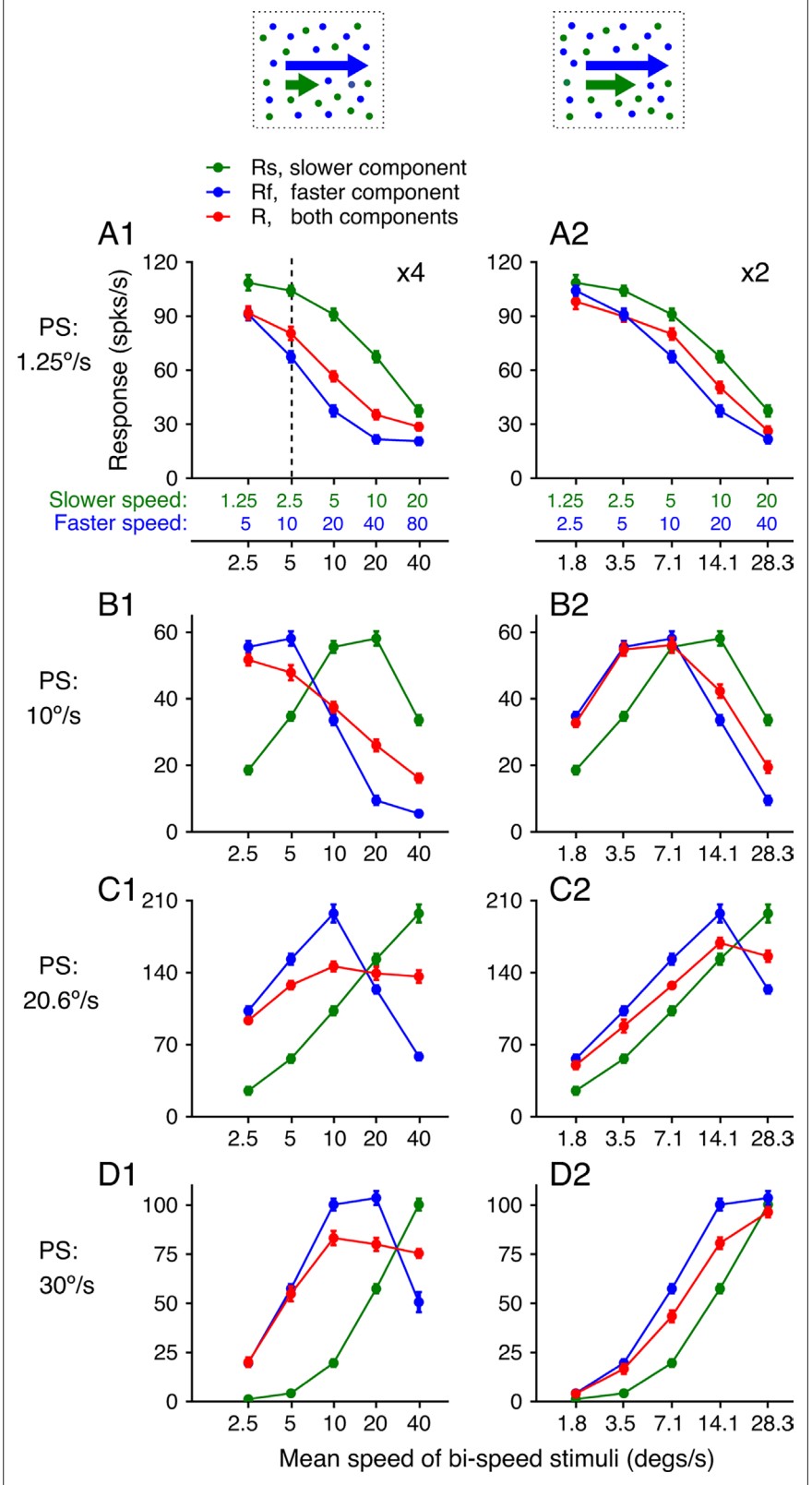

**Figure 3.** Speed tuning curves of four example neurons to bi-speed stimuli and constituent single-speed components. (**A**) Illustration of the visual stimuli and the response tuning curves of an example neuron. Green and blue dots in the diagram indicate two overlapping achromatic random-dot patterns moving in the same direction at different speeds. Colors are used for illustration purposes only. The abscissas in green and blue show the speeds

*Figure 3 continued*

of the slower and faster components, respectively. The abscissa in black shows the log mean speed of the two speed components. (**A–D**) Four example neurons are sorted by their preferred speeds (PS) from slow to fast. Error bars represent ±STE. For some data points, error bars were comparable to the symbol size. (**A1–D1**) 4x speed separation. (**A2–D2**) 2x speed separation.

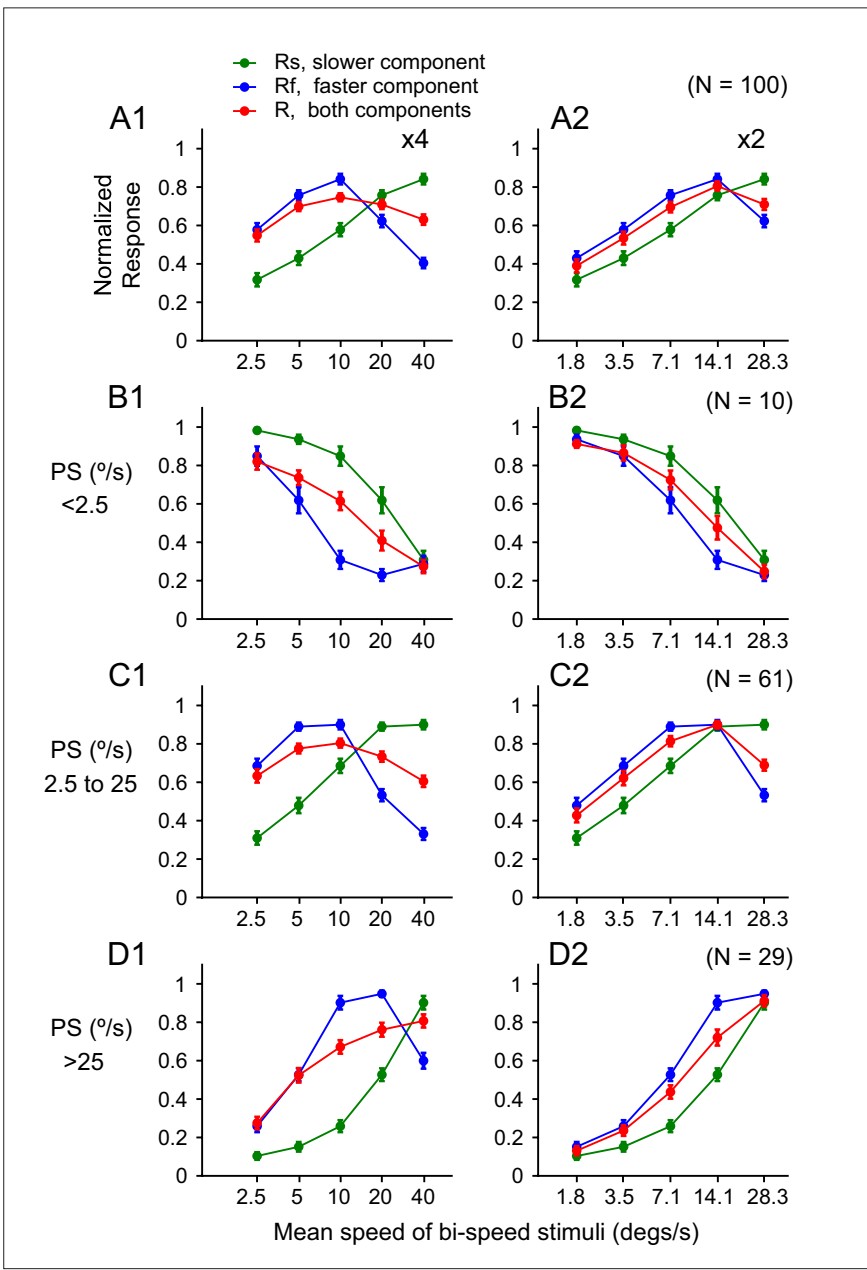

**Figure 4.** Population-averaged speed tuning curves to bi-speed stimuli and constituent single-speed components. Speed tuning curves averaged across: (**A**) 100 neurons in our dataset. (**B**) 10 neurons that had preferred speeds (PS) lower than 2.5°/s. (**C**) 61 neurons that had PS between 2.5 and 25°/s. (**D**) 29 neurons that had PS greater than 25°/s. Error bars represent ± STE. For some data points, error bars were comparable to the symbol size. (**A1–D1**) 4x speed separation. (**A2–D2**) 2x speed separation.

in which, $w_s$ and $w_f$ are the response weights for the slower and faster speed component $v_s$ and $v_f$, respectively.

Our goal was to estimate the weights for each speed pair and determine whether the weights change with the stimulus speeds. In our main data set, the two speed components moved in the same direction. To determine the weights of $w_s$ and $w_f$ for each neuron at each speed pair, we have three data points $R$, $R_s$, and $R_f$, which are trial-averaged responses. Since it is not possible to solve for both variables, $w_s$ and $w_f$, from a single equation *Equation 1* with three data points, we introduced an additional constraint: $w_s + w_f = 1$. With this constraint, the weighted sum becomes a weighted average. While this constraint may not yield the exact weights that would be obtained with a fully determined system, it nevertheless allows us to characterize how the relative weights vary with stimulus speed. As long as $R_f \neq R_s$, $R$ can be expressed as:

$$R = \frac{R_f - R}{R_f - R_s} R_s + \frac{R - R_s}{R_f - R_s} R_f, \tag{2}$$

The response weights are $w_s = \frac{R_f - R}{R_f - R_s}$, $w_f = \frac{R - R_s}{R_f - R_s}$. Intuitively, if $R$ were closer to one component response, that stimulus component would have a higher weight. Note that *Equation 2* is not intended for fitting the response $R$ using $R_s$ and $R_f$, but rather to use the relationship among $R$, $R_s$, and $R_f$ to determine the weights for the faster and slower components.

Using this approach to estimate response weights for individual neurons can be unreliable, particularly when $R_f$ and $Rs$ are similar. This situation often arises when the two speeds fall on opposite sides of the neuron's preferred speed, resulting in a small denominator ($R_f - Rs$) and consequently an artificially inflated weight estimate. We, therefore, used the neuronal responses across the population to determine the response weights (*Figure 5*). For each pair of stimulus speeds, we plotted ($R-R_s$) in the ordinate versus ($R_f - R_s$) in the abscissa. *Figure 5A1–E1* shows the results obtained at 4x speed separation. Across the neuronal population, the relationship between ($R - R_s$) and ($R_f - R_s$) can be described by a linear equation (*Equation 3*) (see $R^2$ in *Table 1*). This linearity suggests that the response weights for each speed pair are roughly consistent across the neuronal population.

$$R - R_s = k\left(R_f - R_s\right) + b \tag{3}$$

Because all the regression lines in *Figure 5* nearly go through the origin (i.e. intercept $b \approx 0$, *Table 1*), the slope $k$ obtained from the linear regression approximates $\frac{R - R_s}{R_f - R_s}$, which is the response weight $w_f$ for the faster component (*Equation 2*). Hence, for each pair of stimulus speeds, we can estimate the response weight for the faster component using the slope of the linear regression of the responses from the neuronal population.

Our results showed that the bi-speed response showed a strong bias toward the faster component when the speeds were slow and changed progressively from a scheme of 'faster-component-take-all' to 'response-averaging' as the speeds of the two stimulus components increased (*Figure 5F1*). We found similar results when the speed separation between the stimulus components was small (2x), although the bias toward the faster component at low stimulus speeds was not as strong as 4x speed separation (*Figure 5A2–F2* and *Table 1*).

In the regression between $\left(R - R_s\right)$ and $\left(R_f - R_s\right)$, $R_s$ (i.e. the firing rate to the slow component averaged across all trials for each neuron) was a common term and, therefore, could artificially introduce correlations. We wanted to determine whether our estimates of the regression slope ($w_f$) were confounded by this factor. We performed two additional analyses.

First, at each speed pair and for each of the 100 neurons in the data sample shown in *Figure 5*, we simulated the response to the bi-speed stimuli ($R_e$) as a randomly weighted average of $R_f$ and $R_s$ of the same neuron.

$$R_e = aR_f + \left(1 - a\right)R_s, \tag{4}$$

in which $a$ was a randomly generated weight (between 0 and 1) for $R_f$, and the weights for $R_f$ and $R_s$ summed to one. We then calculated the regression slope and the correlation coefficient between the simulated $R_e - R_s$ and $R_f - R_s$ across the 100 neurons. We repeated the process 1000 times and obtained the mean and 95% confidence interval (CI) of the regression slope and the $R^2$. The mean slope based on the simulated responses was 0.5 across all speed pairs. The estimated slope ($w_f$)

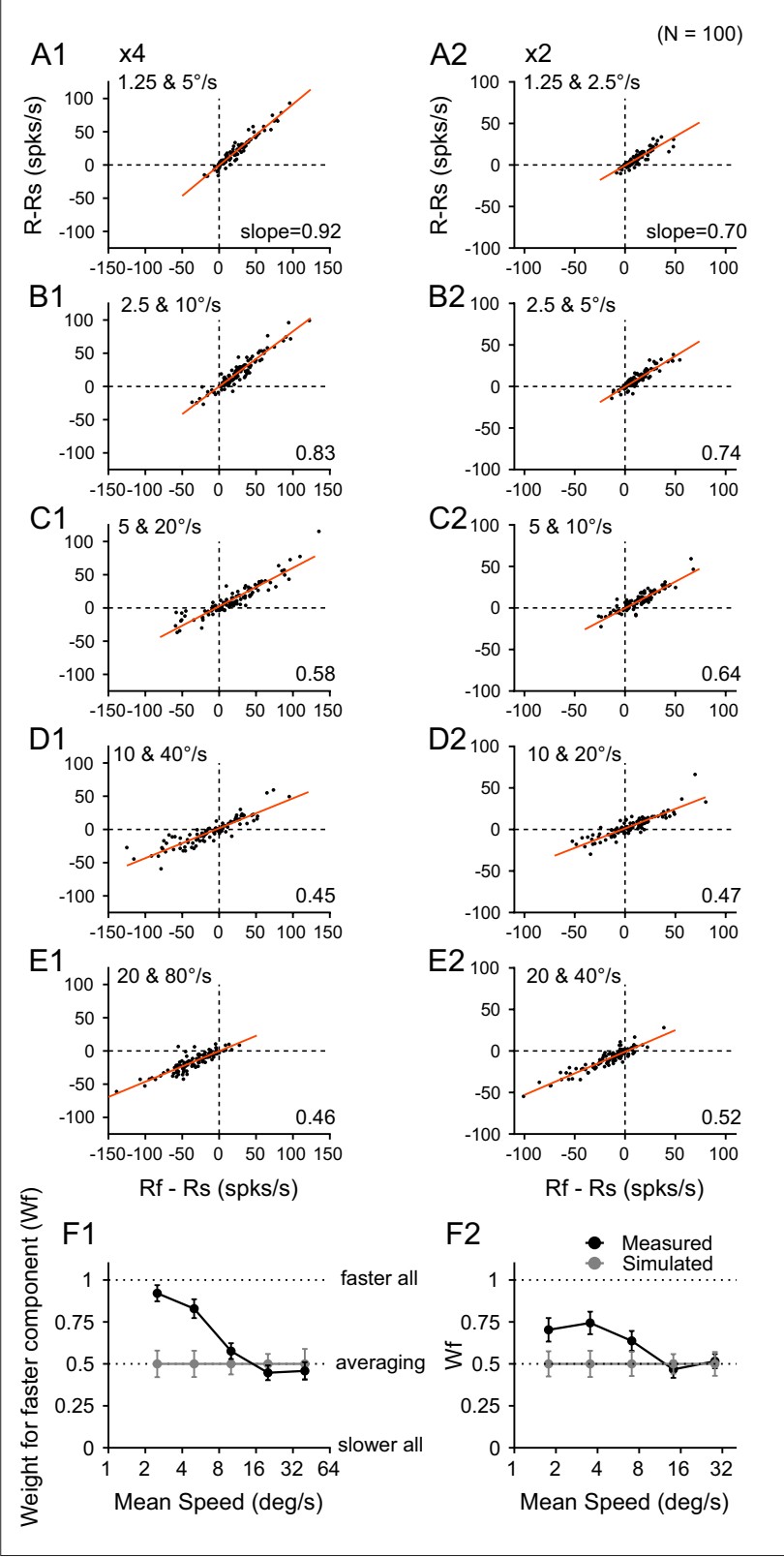

**Figure 5.** Relationship between the responses to the bi-speed stimuli and the constituent stimulus components. (**A–E**) Each panel shows the responses from 100 neurons. Each dot represents the responses from one neuron. $R$, $R_f$, and $R_s$ were firing rates averaged across all recorded trials for each neuron. The ordinate shows the difference between the responses to a bi-speed stimulus and the slower component ($R - R_s$). The abscissa shows the

*Figure 5 continued*

difference between the responses to the faster and slower components ($R_f$ - $R_s$). The regression line is shown in red. (**F**) Response weights for the faster stimulus component obtained from the slope of the linear regression based on the recorded responses of 100 neurons (black symbols), and based on simulated responses to the bi-speed stimuli (gray symbols). Error bars represent 95% confidence intervals. (**A1–F1**) 4x speed separation. (**A2–F2**) 2x speed separation.

from the data was significantly greater than the simulated slope at slow speeds of 1.25/5, 2.5/10 (*Figure 5F1*), and 1.25/2.5, 2.5/5, and 5/10°/s (*Figure 5F2*) (bootstrap test, see p-values in *Table 1*). The estimated $R^2$ based on the data was also significantly higher than the simulated $R^2$ for most of the speed pairs (*Table 1*).

Second, we calculated $R_s$ in the ordinate and abscissa of *Figure 5A–E* using responses averaged across different subsets of trials, such that $R_s$ was no longer a common term in the ordinate and abscissa. For each neuron, we determined $R_{s1}$ by averaging the firing rates of $R_s$ across half of the recorded trials, selected randomly. We also determined $R_{s2}$ by averaging the firing rates of $R_s$ across the rest of the trials. We regressed $(R - R_{s1})$ on $(R_f - R_{s2})$, as well as $(R - R_{s2})$ on $(R_f - R_{s1})$, and repeated the procedure 50 times. The averaged slopes obtained with $R_s$ from the split trials showed the same pattern as those using $R_s$ from all trials (*Table 1* and *Appendix 1—figure 1*), although the coefficient of determination was slightly reduced (*Table 1*). For 4x speed separation, the slopes were nearly identical to those shown in *Figure 5F1*. For 2x speed separation, the slopes were slightly smaller than those in *Figure 5F2*, but followed the same pattern (*Appendix 1—figure 1*). Together, these analysis results confirmed the faster-speed bias at the slow stimulus speeds and the change of the response weights as stimulus speeds increased.

**Table 1.** Response weight for faster component based on linear regression (N=100).

| | Large speed difference (4x) | | | | | Small speed difference (2x) | | | | |
|---|---|---|---|---|---|---|---|---|---|---|
| Components speeds (°/s) | 1.25/5 | 2.5/10 | 5/20 | 10/40 | 20/80 | 1.25/2.5 | 2.5/5 | 5/10 | 10/20 | 20/40 |
| Intercept (*b*) | −0.60 | −0.13 | 2.34 | 1.79 | −0.33 | −0.65 | −0.45 | −0.32 | 1.23 | −0.99 |
| Slope ($w_f$) and 95% CI | 0.92 ± 0.048 | 0.83 ± 0.056 | 0.58 ± 0.047 | 0.45 ± 0.044 | 0.46 ± 0.052 | 0.70 ± 0.070 | 0.74 ± 0.067 | 0.64 ± 0.059 | 0.47 ± 0.050 | 0.52 ± 0.042 |
| Simulated slope ($w_f$) and 95% CI | 0.50 ± 0.079 | 0.50 ± 0.078 | 0.50 ± 0.063 | 0.50 ± 0.059 | 0.50 ± 0.089 | 0.50 ± 0.075 | 0.50 ± 0.078 | 0.50 ± 0.072 | 0.50 ± 0.058 | 0.50 ± 0.071 |
| p-values ($w_f$) (*measured*>simulated) | <0.001 (***) | <0.001 (***) | 0.09 | 0.86 | 0.686 | 0.005 (**) | 0.002 (**) | 0.017 (*) | 0.742 | 0.432 |
| $R^2$ | 0.94 | 0.90 | 0.86 | 0.80 | 0.76 | 0.80 | 0.83 | 0.82 | 0.78 | 0.86 |
| Simulated $R^2$ and 95% CI | 0.62 ± 0.162 | 0.62 ± 0.165 | 0.71 ± 0.111 | 0.73 ± 0.095 | 0.55 ± 0.176 | 0.64 ± 0.159 | 0.62 ± 0.158 | 0.66 ± 0.137 | 0.75 ± 0.098 | 0.66 ± 0.154 |
| p-values ($R^2$) (measured > simulated) | <0.001 (***) | <0.001 (***) | <0.001 (***) | 0.096 | 0.003 (**) | 0.01 (**) | 0.003 (**) | <0.001 (***) | 0.311 | 0.002 (**) |
| Slope ($w_f$) ± STD ($R_s$ from split trials) | 0.90 ± 0.021 | 0.81 ± 0.020 | 0.56 ± 0.015 | 0.44 ± 0.015 | 0.44 ± 0.024 | 0.63 ± 0.075 | 0.67 ± 0.078 | 0.58 ± 0.072 | 0.44 ± 0.058 | 0.48 ± 0.071 |
| $R^2$ ($R_s$ from split trials) | 0.89 | 0.85 | 0.82 | 0.75 | 0.67 | 0.63 | 0.65 | 0.66 | 0.66 | 0.73 |

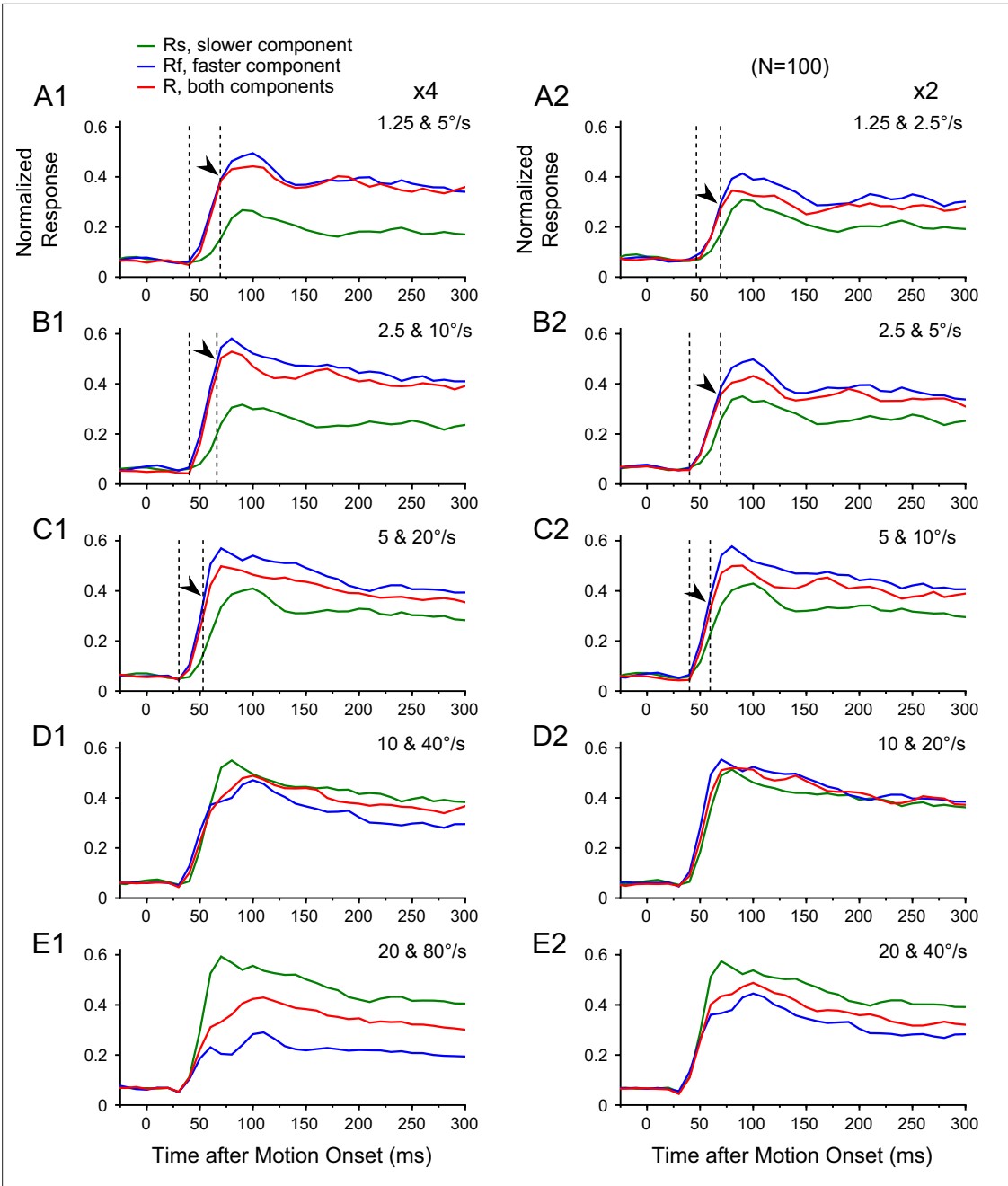

**Figure 6.** Timecourse of MT responses averaged across neurons to bi-speed stimuli. Peristimulus time histograms (PSTHs) were averaged across 100 neurons. The bin width of the PSTH was 10 ms. (**A1–E1**) 4x speed separation. (**A2–E2**) 2x speed separation. In A-C, the left dashed line indicates the latency of the response to a bi-speed stimulus, and the right dashed line and the arrow indicate when the response to a bi-speed stimulus started to diverge from the response to the faster component.

## Timecourse of MT responses to bi-speed stimuli

The temporal dynamics of the response bias toward the faster component may provide a useful constraint on the neural model that accounts for this phenomenon. We, therefore, examined the timecourse of MT response to the bi-speed stimuli. We asked whether the faster-speed bias occurred early in the neuronal response or developed gradually.

*Figure 6* shows the timecourse of the normalized responses averaged across 100 neurons in the population. The bias toward the faster speed component occurred at the very beginning of the neuronal response when the stimulus speeds were less than 20°/s (*Figure 6A–C*). The first 20–30

ms of the neuronal response elicited by the bi-speed stimulus was nearly identical to the response elicited by the faster component alone, as if the slower component were not present. The early dominance of the faster component on the bi-speed response cannot be explained by the difference in the response latencies of the faster and slower components. Faster stimuli elicit a shorter response latency (*Lisberger and Movshon, 1999*), which can be seen in *Figure 6A–C*. However, the bi-speed response still closely followed the faster component for some time after the response to the slower component began to rise. The effect of the slower component on the bi-speed response was delayed for about 25 ms, as indicated by the arrows in *Figure 6A–C*. During the rest of the response period, the bias toward the faster component was persistent. As the stimulus speeds increased, the bi-speed response gradually changed to follow the average of the component responses (*Figure 6E*). We found similar results when the speed separation between the two stimulus components was 4x (*Figure 6A1– E1*) and 2x (*Figure 6A2–E2*). At slow speeds, the very early faster-speed bias suggests a likely role of feedforward inputs to MT in the faster-speed bias. The slightly delayed reduction (normalization) in the bi-speed response relative to the stronger component response also helps constrain the circuit model for divisive normalization.

## Faster-speed bias is still present when attention was directed away from the RFs

One possible explanation for the faster-speed bias is that bottom-up attention is drawn toward the faster stimulus component, enhancing the response to it. To address this question, we asked whether the faster-speed bias was still present if attention was directed away from the RFs. We trained one monkey (RG) to perform a demanding direction-discrimination task in the visual hemifield opposite to the RFs. The monkey performed the task well with an average correct rate of 86.7 ± 7.3% (mean ± std) (see Methods and *Appendix 1—figure 2*).

We recorded the responses from an additional 48 MT neurons in 23 experimental sessions while the monkey performed the task. Thirty-two out of the 48 neurons were recorded using both the attention-away paradigm and a fixation paradigm. The results obtained using the attention-away paradigm and the fixation paradigm were similar (*Appendix 1—figure 2*). The faster-speed bias was more evident at 4x speed separation than at 2x speed separation. Based on the neuronal responses across the population, we calculated the weight for the faster stimulus component at each of the five speed pairs using linear regression *Equation 2; Equation 3*, as we did in *Figure 5*. When attention was directed away from the RFs, the response weight for the faster component decreased from a strong faster-speed bias to response averaging as the stimulus speeds increased, similar to the results from the fixation paradigm (*Figure 7*). These results suggest that the faster-speed bias at low speeds cannot be explained by attention drawn to the faster-speed component.

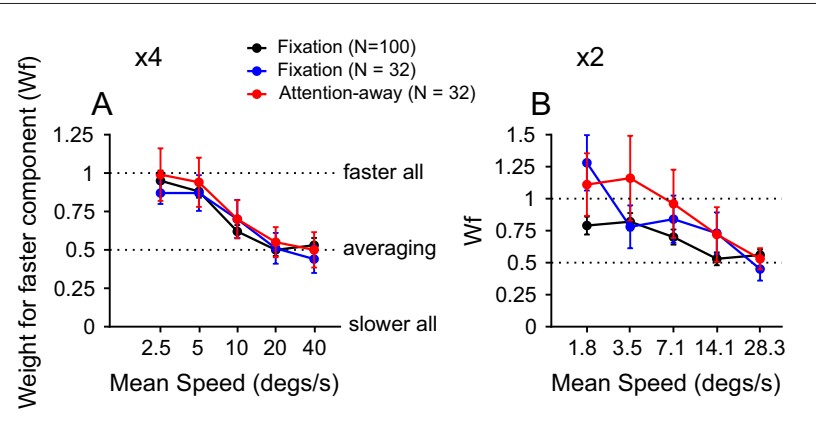

**Figure 7.** Comparison of response weights between attention-away and fixation paradigms. The red and blue curves indicate the response weights for the faster speed component in an attention-away paradigm and a fixation paradigm, respectively, obtained from the same population of 32 neurons. The black curves are the replot of the data in *Figure 5F*, obtained from 100 neurons in a fixation paradigm. (**A**) 4x speed separation. (**B**) 2x speed separation.

## Faster speed bias also occurs when stimulus components move in different directions

We showed that at low speeds, MT response to the bi-speed stimulus was biased toward the faster stimulus component when two overlapping components moved in the same direction (at the preferred direction of the neuron). We asked whether this faster-speed bias also occurred when visual stimuli moved in different directions. We presented overlapping random-dot stimuli within the RF, moving in two directions separated by 90°. The two stimulus components moved at speeds of 2.5 and 10°/s. The faster speed component moved on the clockwise side of the two directions. We varied the vector-average (VA) direction of the two component directions across 360° to characterize the direction tuning curves. Each neuron's direction tuning curve was fitted with a spline and circularly shifted such that the VA direction 0° was aligned with the neuron's preferred direction before averaging across neurons.

*Figure 8A* shows the results averaged across 21 neurons (13 from monkey RG, 8 from monkey GE). The peak response to the faster component (*Figure 8A*, blue curve) was stronger than that to the slower component (green curve), consistent with the overall speed preference of a large MT neuron population (*Nover et al., 2005*). MT responses elicited by the bi-directional stimuli (red curve) showed a strong bias toward the faster component, more than expected by the average of the two component responses (gray curve). The bi-speed response was biased toward the faster component regardless of whether the response to the faster component was stronger (in positive VA directions) or weaker (in negative VA directions) than that to the slower component (*Figure 8A*). The result from an example neuron further demonstrated that, even when the peak firing rates of the faster and slower component responses were similar, the response elicited by the bi-speed stimuli was still biased toward the faster component (*Figure 8B*). These results suggest that the bias was not toward the stronger component response of the individual neuron, but to the faster component.

To quantify the response weights, for each neuron, we fitted the MT raw firing rates of the direction tuning curve to bi-speed/bi-directional stimuli as a linear weighted sum (LWS) of the direction tuning curves to the individual stimulus components moving at different speeds:

$$R_{bi}\left(\theta_1, \theta_2\right) = w_s \bullet R_s\left(\theta_1\right) + w_f \bullet R_f\left(\theta_2\right) + c. \tag{5}$$

$R_{bi}$ is the model-fitted direction-tuning curves to the bi-speed and bi-direction stimuli. $R_s$ and $R_f$ are the measured direction tuning curves to the slower and faster stimulus components, respectively. $\theta_1 \, and \, \theta_2$ are the motion directions of the two components; $w_s$, $w_f$, and $c$ are model parameters, which represent the response weights for the slower and faster components and an offset constant, respectively. $c$ was constrained to be between 0 and 100 spikes/s. Because the model can be well constrained by the measured direction-tuning curves, it is not necessary to require $w_s$ and $w_f$ to sum to one, which is more general. An implicit assumption of the model is that, at a given pair of stimulus speeds, the response weights for the slower and faster components are fixed across motion directions. The model fitted MT responses very well, accounting for an average of 91.8% of the response variance (std = 7.2%, N=21) (see Methods), which supports the assumption that the response weights are fixed across motion directions. The median response weights for the faster and slower components were 0.74 and 0.26, respectively, and were significantly different (Wilcoxon signed-rank test, $p=8.0 \times 10^{-5}$). For most neurons (20 out of 21), the response weight for the faster component was larger than that for the slower component (*Figure 8C*). This result suggests that at low speeds, the faster-speed bias is a general phenomenon that applies to overlapping stimuli moving either in the same direction or different directions.

## Normalization model fit of the direction-tuning curves to bi-speed stimuli

We showed that the neuronal response in MT to a bi-speed stimulus can be described by a weighted sum of the neuron's responses to the individual speed components. However, what determines the response weights? The divisive normalization model (*Carandini and Heeger, 2012*) has been used to explain a wide array of phenomena, including neuronal responses elicited by multiple visual stimuli (e.g. *Britten and Heuer, 1999*; *Heuer and Britten, 2002*; *Busse et al., 2009*; *Xiao et al., 2014*; *Xiao and Huang, 2015*; *Bao and Tsao, 2018*; *Cherian and Maunsell, 2025*; *Wiesner et al., 2025*). In the

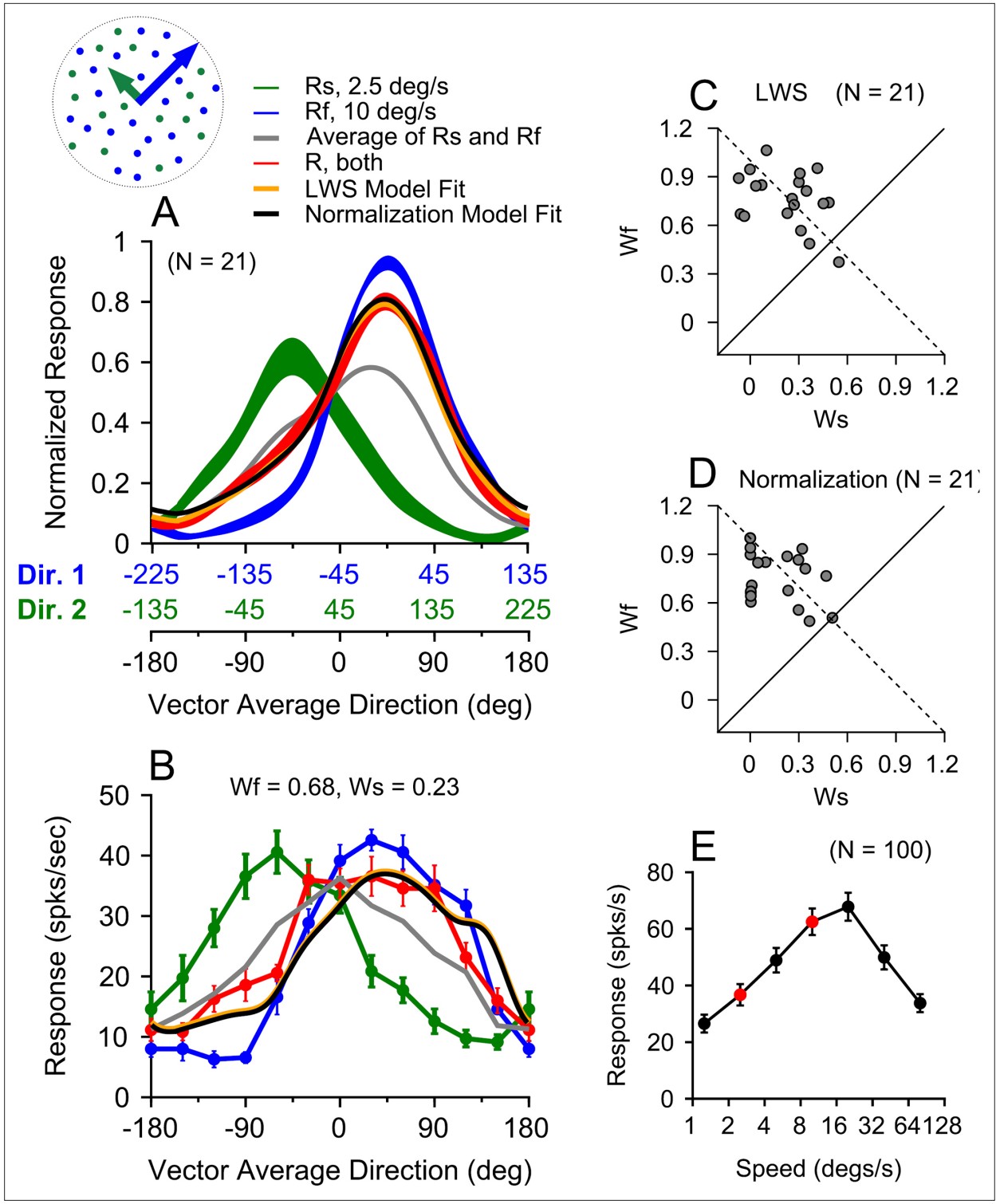

**Figure 8.** MT responses to bi-speed stimuli moving in different directions and the linear weighted sum (LWS) and normalization model fits. (**A**) Population-averaged direction tuning curves of 21 neurons in response to stimuli moving at two speeds and in two directions separated by 90° (red). The component direction Dir. 1 (blue) moved at 10/s, and the component direction Dir. 2 (green) moved at 2.5°/s. The faster component, Dir. 1, was always on the clockwise side of Dir. 2. The abscissas in blue and green show the directions of stimulus components Dir. 1 and Dir. 2, respectively. The blue and green axes are shifted by 90° relative to each other. The abscissa in black shows the corresponding vector-average (VA) direction of the two direction components. Error bands represent ± STE. The gray curve represents the average of the component responses. The orange and black curves are the linear weighted sum (LWS) and normalization model fits, respectively, of the population-averaged direction-tuning curve to the bi-speed stimuli.

*Figure 8 continued on next page*

*Figure 8 continued*

(**B**) The direction-tuning curves of an example neuron show similar peak responses to the slower and faster components. The orange and black curves are the LWS and normalization model fits of the bi-speed responses and are nearly identical. The weights of $w_f$, $w_s$ are from the normalization model fit. (**C**) Response weights for the stimulus components obtained using the LWS model fit. Each circle represents one neuron. (**D**) Response weights obtained using the normalization model fit. The dashed lines in C, D indicate where $w_s$ and $w_f$ sum to one. Although $w_s$ and $w_f$ are not constrained to sum to one in the model fits, the fitted weights are roughly aligned with the dashed lines. (**E**) Population-averaged speed tuning curves of MT neurons recorded in our data sample in response to single speeds. The red circles indicate responses to 2.5 and 10°/s. Error bars represent ± STE.

normalization model, while the division by the activity of a population of neurons in the denominator (the normalization pool) is well accepted, the nature of the numerator is less understood. We have previously proposed that the weight of a stimulus component is proportional to the activity of a population of neurons elicited by the stimulus component (**Xiao et al., 2014**; **Wiesner et al., 2020**). We refer to this neuronal population as the 'weighting pool.' Here, we assumed that the weighting pool was composed of neurons with a broad range of speed preferences in response to multiple speed components. So, the summed response of the weighting pool reflects the speed preference of the neuronal population rather than the speed preference of individual neurons. We used the following equation (**Equation 6**) to fit the direction-tuning curves of each neuron in response to two speed components moving in different directions:

$$R_{bi}\left(\theta_1, \theta_2\right) = \frac{S_s^n}{S_s^n + \alpha S_f^n + \sigma} \bullet R_s\left(\theta_1\right) + \frac{S_f^n}{S_s^n + \alpha S_f^n + \sigma} \bullet R_f\left(\theta_2\right) + c. \tag{6}$$

$R_{bi}$, $R_s$, $R_f$, $\theta_1$, and $\theta_2$ are the same as in **Equation 5**. $S_s$ and $S_f$ are the population neural responses of the weighting pools to the slower and faster component speeds, respectively. $n$, $\sigma$, $\alpha$, and $c$ are model parameters and have the following constraints: $0.01 \leq n \leq 100$, $0 \leq \sigma \leq 500$, $0.01 \leq \alpha \leq 100$, $0 \leq c \leq 100$. $\alpha$ is a parameter that controls for the tuned normalization (**Ni et al., 2012**; **Rust et al., 2006**; **Carandini et al., 1997**). We approximated $S_s$ and $S_f$ based on the population-averaged responses of our recorded MT neurons (N=100) in response to single speeds moving in the preferred direction of each neuron (**Figure 8E**). For the speeds of 2.5 and 10°/s, $S_s$ = 36.7 and $S_f$ = 62.5 spikes/s (red circles in **Figure 8E**). The normalization model fitted the data well, accounting for an average of 90.5% of the response variance (std = 7.1%, N=21), slightly smaller but comparable to the fit by the LWS model. The median response weights obtained from the normalization model for the faster and slower components were 0.78 and 0.15, respectively, and were significantly different (Wilcoxon signed-rank test, $p$=6.0 × 10$^{-5}$) (**Figure 8D**). The median values of the fitted parameters across 21 neurons are n=4.13, σ=123, $\alpha$=1.57, c=0.03.

So far, we have described the neural encoding of multiple speeds in area MT. We will next examine the decoding of speed(s) from population neural responses in MT and compare the performance of decoding with perceptual performance.

## Discriminate bi-speed and single-speed stimuli based on neuronal responses in area MT

We asked whether the responses of MT neurons contained information about bi-speed and single-speed stimuli suitable for supporting the perceptual discrimination of these stimuli. To address this question, we first examined the responses elicited by the bi-speed and single-speed stimuli from a population of MT neurons with different preferred speeds. Next, we used a classifier to discriminate the bi-speed stimuli from the single, log-mean speed stimuli based on MT responses.

In different experimental sessions, we centered visual stimuli on neurons' RFs. The visual stimuli were identical across experimental sessions except for the spatial location of the RF. This allowed us to pool the trial-averaged responses recorded from different neurons to form a pseudo-population (see Methods). One can interpret the responses as from a population of neurons elicited by the same visual stimulus. **Figure 9** shows the pseudo-population neural response (referred to in brief as the population response) plotted as a function of neurons' preferred speed, constructed from 100 neurons that we recorded using a fixation paradigm (see Methods). To capture the population response evenly across a full range of preferred speeds, we spline-fitted the recorded response elicited by the bi-speed stimulus (the red curves) and by the single, log-mean speed (the black curves) (**Figure 9A–E**). At 4x and 2x

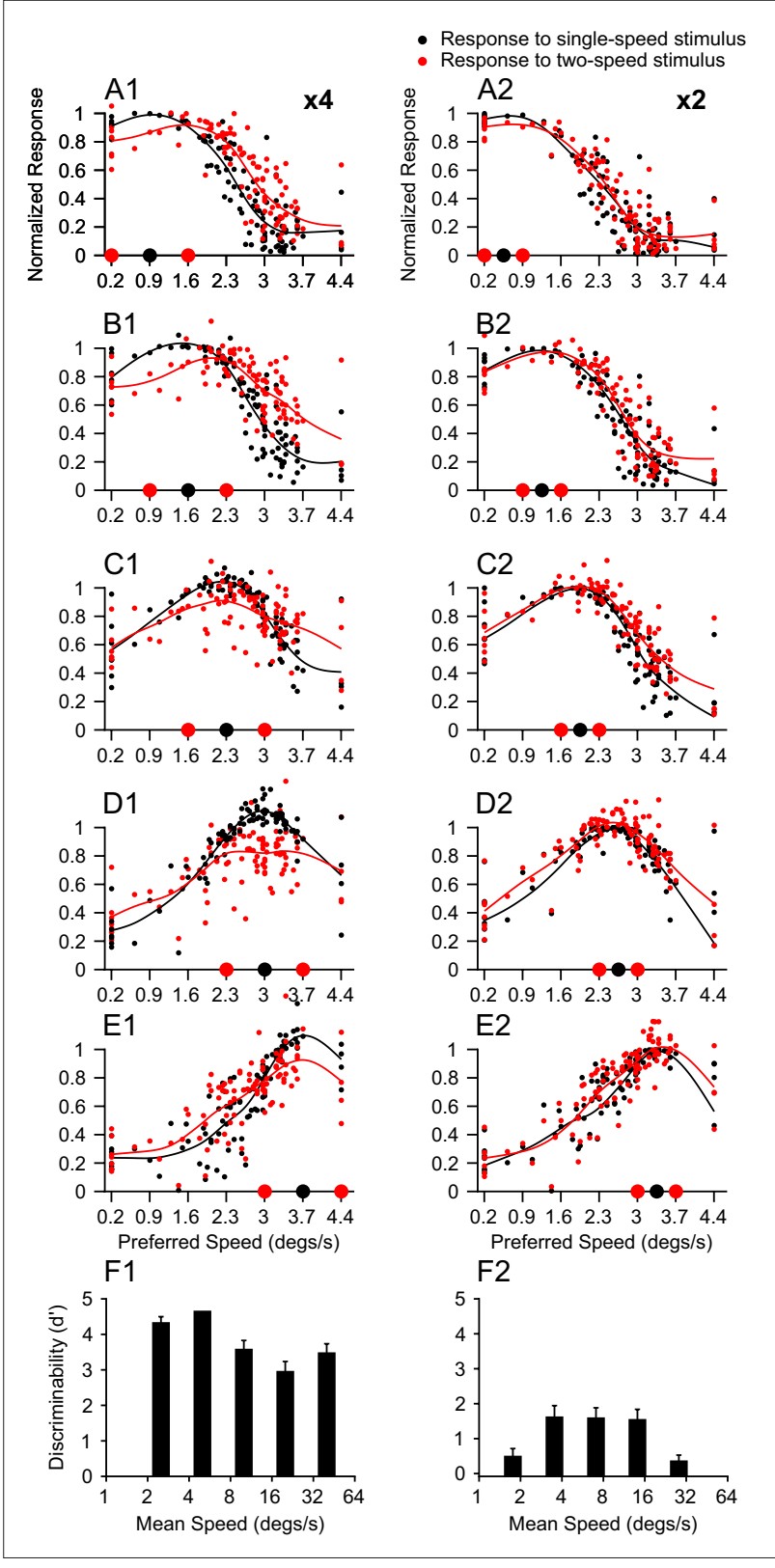

**Figure 9.** Population neural responses elicited by the bi-speed and single-speed stimuli and the performance of a linear classifier. A population response of 100 recorded neurons was reconstructed by pooling across recordings in different experimental sessions. Each neuron's response was averaged across experimental trials and normalized by the maximum response of the spline-fitted speed tuning curve to single speeds. Each dot represents the

*Figure 9 continued*

response from one neuron plotted as the neuron's preferred speed (PS) in the natural logarithm scale. The curves represent the spline-fitted population neural responses. Red: response to the bi-speed stimulus; Black: the response to the corresponding single, log-mean speed. (**A1–F1**) 4x speed separation. The speeds of the bi-speed stimuli are 1.25 and 5°/s (**A1**), 2.5 and 10°/s (**B1**), 5 and 20°/s (**C1**), 10 and 40°/s (**D1**), 20 and 80°/s (**E1**). (**A2–F2**) 2x speed separation. The speeds of the bi-speed stimuli are 1.25 and 2.5°/s (**A2**), 2.5 and 5°/s (**B2**), 5 and 10°/s (**C2**), 10 and 20°/s (**D2**), 20 and 40°/s (**E2**). Two red dots on the X-axis indicate two component speeds; the black dot indicates the log-mean speed. (**F1, F2**) Performance of a linear classifier to discriminate the population neural responses to the bi-speed stimulus and the corresponding single log-mean speed. Error bars represent STE.

speed separations, the population responses elicited by two speeds did not show two separate peaks. Instead, they had a single hump that shifted from low to high preferred speed as the stimulus speeds increased. At 4x speed separation across all five speed pairs, the population response elicited by two speeds was broader and flatter than that elicited by the single log-mean speed (*Figure 9A1–E1*).

In our experiments, we directly measured the neuronal responses elicited by the log-mean speed of 4x but not 2x speed separation. Because we had characterized each neuron's tuning curve to single speeds, we could infer the responses elicited by the log-mean speed of 2x separation by interpolating the speed tuning curve using a spline fit. At 2x speed separation, the population response elicited by two speeds was similar to that elicited by the single log-mean speed, with the two-speed population response slightly broader (*Figure 9A2–E2*).

We used a linear classifier to perform a discrimination task to evaluate the discriminability between MT population responses elicited by the bi-speed stimulus and the corresponding log-mean speed. Trial-by-trial population responses were generated randomly according to a Poisson process, and the mean response of each neuron was set to the trial-averaged neuronal response. The classifier was trained and tested using k-fold cross-validation. The classifier determined whether a population response from the recorded 100 neurons in our data set was elicited by two speeds or a single speed (see Methods). Discriminability of the classifier was measured in *d'* as in our psychophysics study.

Consistent with perceptual discrimination (*Figures 1E and 2B*), the classifier's performance at 4x speed separation (*Figure 9F1*) was better than that at 2x speed separation (*Figure 9F2*). This provides a neural correlate with better perceptual speed segmentation at larger speed separation. At 4x speed separation, the discriminability of the classifier was slightly decreased as the stimulus speed increased (*Figure 9F1*), which was generally consistent with the human psychophysics results (*Figure 1E1*). However, one difference was that at 20 and 80°/s, the classifier's performance did not drop to a low level as human performance (compare *Figure 9F1* with *Figure 1E1*), but was more comparable to that of the monkey subject (*Figure 2B1*). At 2x speed separation, the classifier's performance (*Figure 9F2*) had a similar shape as that of the human (*Figure 1E2*) and monkey (*Figure 2B2*) subjects, but the performance was not as good as the perceptual performance at intermediate speeds.

When the stimulus speeds were 20 and 80°/s, the population responses elicited by the bi-speed stimulus and the single log-mean speed stimulus were noticeably different (*Figure 9E1*), which explains the good performance of the classifier in differentiating the two stimuli. However, the differences in population neural responses may contribute to perceptual differences in quality other than motion speeds, and the monkey subject might be able to detect these perceptual cues at high speeds to aid task performance. To directly evaluate whether the population neural responses elicited by the bi-speed stimulus carry information about two speeds, it is important to conduct a decoding analysis to extract speed(s) from MT population responses.

## Decoding either a single speed or two speeds from trial-averaged population neural response

Since the population responses elicited by the 4x and 2x speed separations only had a single peak centered between the two component speeds (*Figure 9A–E*), this raised the question of how neuronal populations represent multiple speeds of the motion components. To address this question, we used a decoding approach motivated by the theoretical framework of coding multiplicity and probability distribution of visual features in neuronal populations proposed by Zemel et al. (*Zemel et al., 1998*; *Pouget et al., 2003*; also see *Treue et al., 2000*). Our decoder extracted speeds that minimized the difference (sum squared error) between the estimated population response elicited by the extracted

speeds and the reconstructed population neural response based on the neural recording (*Equations 7–10*, see Methods). Rather than searching for a probability distribution of speed, we constrained the search to either a single speed or two speeds. We also constrained the weights for the extracted speeds to sum to one, consistent with a probability distribution.

Our approach is akin to the forward encoding model for decoding that is often used in brain imaging studies (e.g. *Kay et al., 2008*; *Brouwer and Heeger, 2009*; *Naselaris et al., 2011*; *Vintch and Gardner, 2014*; *van Bergen et al., 2015*). We applied an encoding rule and found that the visual stimuli generating a population response best matched the recorded neural response. Our assumed encoding rule in the decoder is that a neuron's response to multiple speeds is the linear sum of the neuron's responses to individual speed components presented alone, based on the neuron's speed tuning curve and weighted by the strength (or probability) of each speed component. The decision to use this encoding model for decoding, rather than the encoding rule characterized in this study, was made primarily for practical reasons. Our experimental data only covered two speed separations (4x and 2x) and 5 log mean speeds. We do not yet know a general encoding rule for two speeds across all different speed separations and log mean speeds. However, if the linear encoding of the two speeds, as characterized in this study, generalizes across a broader range of speed combinations – such that only the weights of the speed components vary within the general encoding rule – then our choice of encoding model for decoding would not alter the decoded speeds themselves, but would merely affect the estimated weights associated with those speeds (*Equation 7*).

*Figure 10* shows the decoding procedure and the results of extracting speed(s) from the population neural responses reconstructed based on the trial-averaged responses of the recorded neurons to the bi-speed stimuli. To capture the population neural response across a full range of preferred speeds, we spline-fitted the recorded (red dots) and estimated (blue dots) population responses. The estimated population responses (*Figure 10*, blue curves) matched the recorded neural responses well (*Figure 10*, red curves) (for five speed pairs, $R^2 > 0.96$ at 4x speed separation; $R^2 > 0.99$ at 2x speed separation). At 4x speed separation, the decoder extracted two speeds for all speed combinations (*Figure 10A–E*). The readout speeds were generally close to the veridical stimulus speeds. At low stimulus speeds of 1.25 and 5°/s (*Figure 10A*) and 2.5 and 10°/s (*Figure 10B*), the decoded faster speed component had a higher weight than the slower component. At the highest speeds of 20 and 80°/s, the decoder extracted two speeds (*Figure 10E*), whereas human subjects could not perceive two speeds (*Figure 1E1*) (see *Appendix 1—figure 3*). At 2x speed separation, the decoder extracted two speeds only at low stimulus speeds of 1.25 and 2.5°/s (*Figure 10F*). At higher stimulus speeds, the decoder extracted a dominant speed that was between the two component speeds, with or without a second nearby speed that had a very low weight (*Figure 10G–J*). In contrast, human subjects could perceive two speeds when stimulus speeds were below 20 and 40°/s (*Figure 1E2*) (see below and Discussion).

## Decoding speeds from trial-by-trial population neural responses

To determine the distribution of the readout speed across trials, we randomly generated 200 trials based on the trial-averaged responses of 100 recorded neurons in our data sample. In each simulated trial, a given neuron's response was determined by a Poisson process, with the mean set to the spike count averaged across the recorded trials. The trial-by-trial response of each neuron was normalized to construct the population response and then spline-fitted for decoding. The speeds extracted from the recorded neural responses to single stimulus speeds (*Appendix 1—figure 3A–G*) and from the inferred responses to the log-mean speed of 2x speed separation (*Appendix 1—figure 3H–L*) generally matched the single stimulus speed well (*Appendix 1—figure 3M*).

*Figure 11* shows the speeds extracted from the neural response to the bi-speed stimuli. The decoder often extracted two speeds across trials. In some trials, the readout of one speed component had a minimal weight. We considered a trial having a 'single' readout speed if the weight difference between the two readout speeds was greater than 0.7 (i.e. the weaker weight <0.15). This usually happened when the readout speed having a minimal weight was either at one of the boundaries of the speed range (i.e. 1.25°/s or 80°/s) or separated from the other readout speed by a large speed separation (27.86x, which was the largest speed separation searched by the algorithm) (see Methods). These small weights were likely artifacts due to the boundaries of the stimulus speeds used in our experiments or the range of speed separation searched by the decoder.

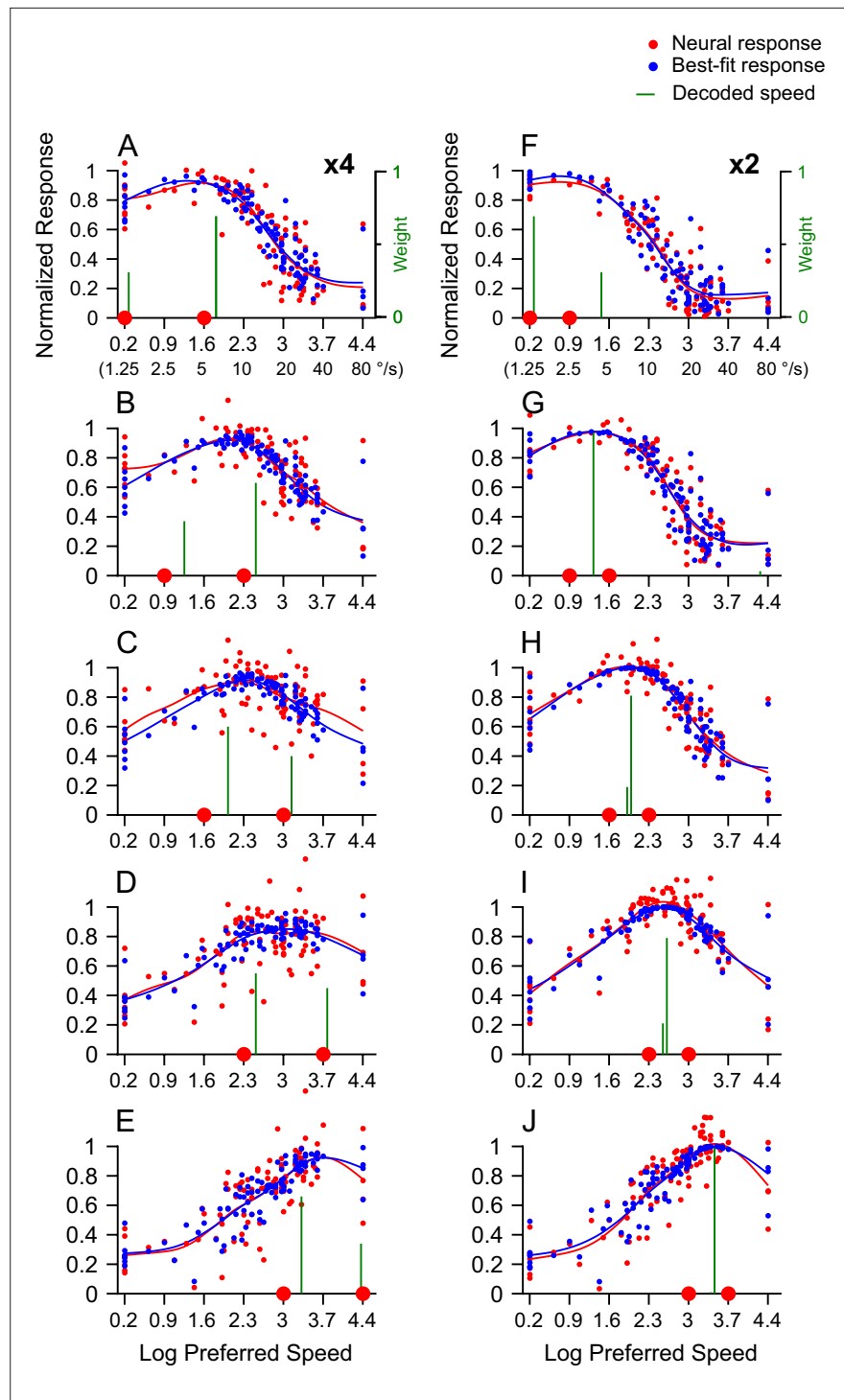

**Figure 10.** Illustration of the decoding procedure and extraction of speed(s) from population responses reconstructed based on the trial-averaged neuronal responses to the bi-speed stimuli. (**A–E**) 4x speed separation. (**F–J**) 2x speed separation. The neural population contains 100 recorded neurons, as shown in *Figure 9*. Each red dot represents the trial-averaged response from one neuron plotted versus the preferred speed (PS) of the neuron in the natural logarithm scale. The red curve represents the spline-fitted population neural response. The decoder found either one speed or two speeds with different weights (vertical green bars on the X-axis), giving rise to the estimated and spline-fitted population response (blue curve) that best fitted the recorded and spline-fitted population neural response (red curve). Each blue dot represents the estimated response from one neuron, and the blue curve represents the spline-fitted estimated population response. Two red dots on the X-axis indicate the stimulus speeds. The Y-axis on the right side shows the weight of the readout speed (**A, F**).

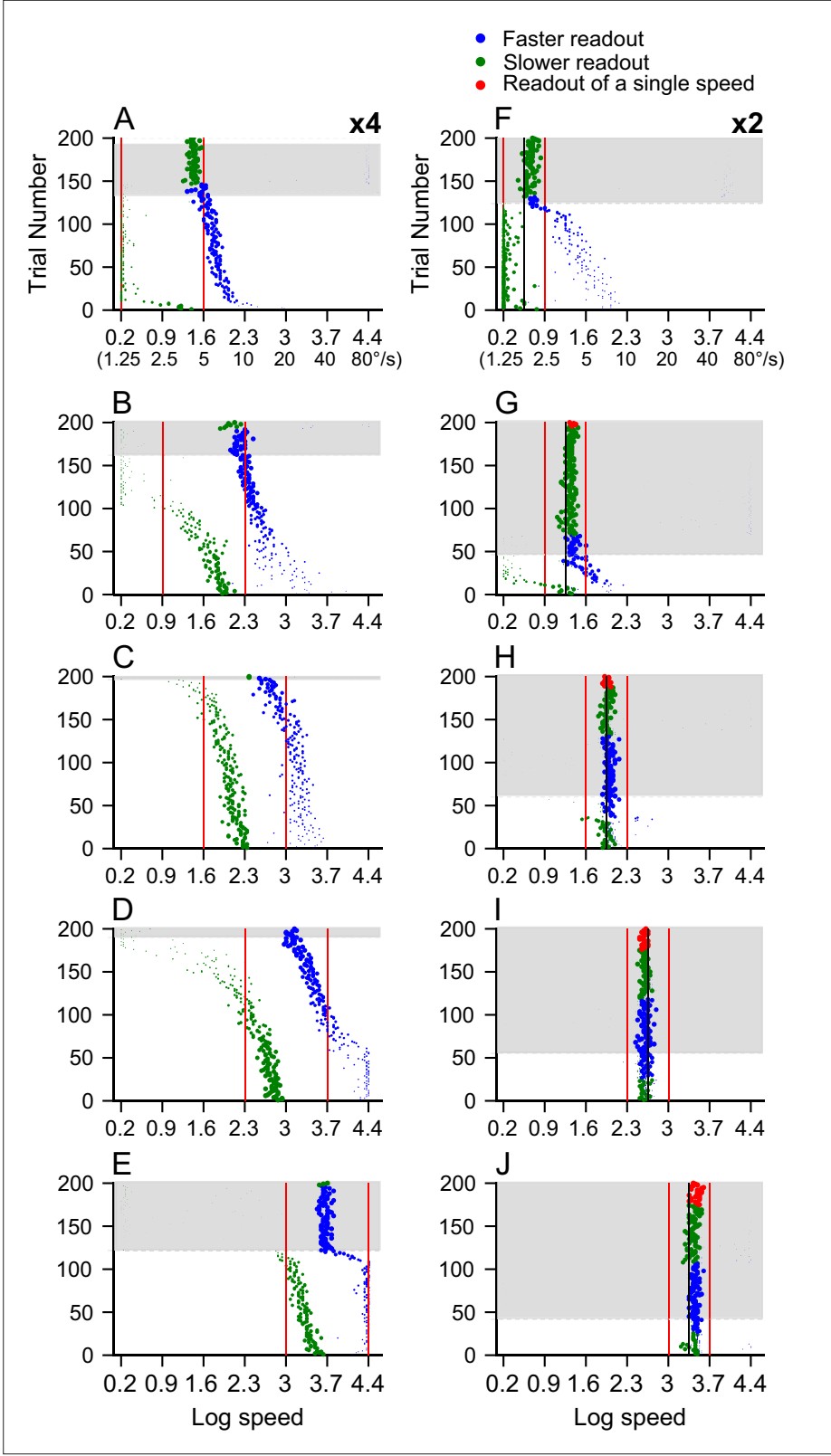

**Figure 11.** Trial-by-trial readout speeds decoded from population neural responses to the bi-speed stimuli. The neural population contains 100 recorded neurons, and the trial-by-trial responses are randomly generated based on a Poisson process. The convention is the same as in *Figure 10*. (A–E) Speeds decoded from population responses to ×4 speed separation. The vertical red lines indicate two component speeds, which are 1.25 and 5⁰/s

*Figure 11 continued on next page*

*Figure 11 continued*

(**A**), 2.5 and 10⁰/s (**B**), 5 and 20⁰/s (**C**), 10 and 40⁰/s (**D**), 20 and 80⁰/s (**E**). (**F–J**) Speeds decoded from population responses to ×2 speed separation. The red vertical line indicates two component speeds, and the black vertical line indicates the log mean speed. The component speeds are 1.25 and 2.5⁰/s (**F**), 2.5 and 5⁰/s (**G**), 5 and 10⁰/s (**H**), 10 and 20⁰/s (**I**), 20 and 40⁰/s (**J**).

At 4x speed separation, the decoder was able to extract the speeds of the stimulus components (*Figure 11A–D*), except at the fastest speeds of 20 and 80°/s. At low stimulus speeds of 1.25 and 5°/s, and 2.5 and 10°/s, the readout speed around the faster stimulus component usually had a higher weight than that around the slower stimulus component (*Figure 11A and B*). At stimulus speeds of 1.25 and 5°/s, in trials with two readout speeds (*Figure 11A*, on the white background), the faster readout speeds were close to the faster stimulus speed of 5°/s. The slower readout speeds were closely aligned with the slower stimulus speed of 1.25°/s, which was also the lower boundary of the speed range. In trials considered to have a single readout speed, the readout was very close to the faster stimulus speed of 5°/s (*Figure 11A*, on the grey background). For some of these trials (at the top of *Figure 11A*), the faster readout speed was near the upper-speed boundary of 80°/s and had a minimal weight (<0.15). Those faster readout speeds were considered boundary artifacts.

At stimulus speeds of 2.5 and 10°/s, the decoder extracted two speeds that had a separation close to the veridical separation (*Figures 11B and 12B*). In trials considered to have a single-speed readout, the readout speed was close to the faster stimulus speed of 10°/s. In some single- and two-readout speed trials, the slower readout speeds aligned with the 1.25°/s boundary and had a small weight, suggesting they were boundary artifacts.

At stimulus speeds of 5 and 20°/s, nearly all trials had two readout speeds with a separation well aligned with the veridical speed separation (*Figures 11C and 12C*). At stimulus speeds of 10 and 40°/s, the decoder was able to extract two speeds for most of the trials (*Figure 11D*). A small percentage of the trials (~10%) were considered to have a single readout speed, which was close to the log mean speed of the two stimulus speeds (20°/s) (blue dots at the top of *Figure 11D* on the grey background).

At the fastest stimulus speeds of 20 and 80°/s, about 40% of the total trials were considered to have only a single readout speed, which was near the log mean speed of the stimulus components (40°/s) (*Figure 11E*). In other trials, the decoder extracted two speeds – the slower readout speeds were generally higher than the slower stimulus speed (20°/s), and the faster readout speeds aligned with the faster stimulus speed (80°/s), which was also the upper boundary speed. However, an examination of the objective function as the decoder searched for the best-fit population response across speed separations revealed that the trial-averaged objective function was flat within a big range of speed separations (*Appendix 1—figure 4A*). Further analysis showed that the decoder was uncertain about how many speeds were in the visual stimuli and, therefore, had difficulty segmenting the visual stimuli at these fast stimulus speeds of 20 and 80°/s (*Appendix 1—figure 4*).

At 2x speed separation, the decoder was not able to extract two speeds of the stimulus components, except at the slowest speeds of 1.25 and 2.5°/s (*Figure 11F–J*). At stimulus speeds of 1.25 and 2.5°/s (*Figure 11F*), in 38% of total trials considered to have a single readout speed, the readout speed was close to the faster stimulus speed of 2.5°/s (mean = 1.97°/s, STD = 1.08). In trials that had two readout speeds, the slower readout speeds roughly followed the slower stimulus speed (1.25°/s), which was also the lower boundary of the speed range (*Figure 11F*). At stimulus speeds higher than 1.25 and 2.5°/s, most trials were considered to have a single readout speed (*Figure 11G–J*). The mean speeds of the single readout-speed trials were 3.9°/s (STD = 1.07), 7.3°/s (STD = 1.99), 13.5°/s (STD = 1.06), and 31°/s (STD = 1.07), respectively, for stimulus speeds of 2.5 and 5°/s, 5 and 10°/s, 10 and 20°/s, and 20 and 40°/s. These mean readout speeds were close to the log mean speeds of the two stimulus speeds (3.54°/s, 7.07°/s, 14.14°/s, and 28.28°/s, respectively).

## Discrimination between single- and bi-speed stimuli based on decoded speeds

To compare the perceptual discrimination between bi-speed stimuli and the log-mean speed, we used the decoding results to perform a discrimination task similar to that used in our psychophysical experiments. *Figure 12* shows the distributions of the speed separation between two readout speeds extracted from the reconstructed population neural responses to the bi-speed stimuli, and the

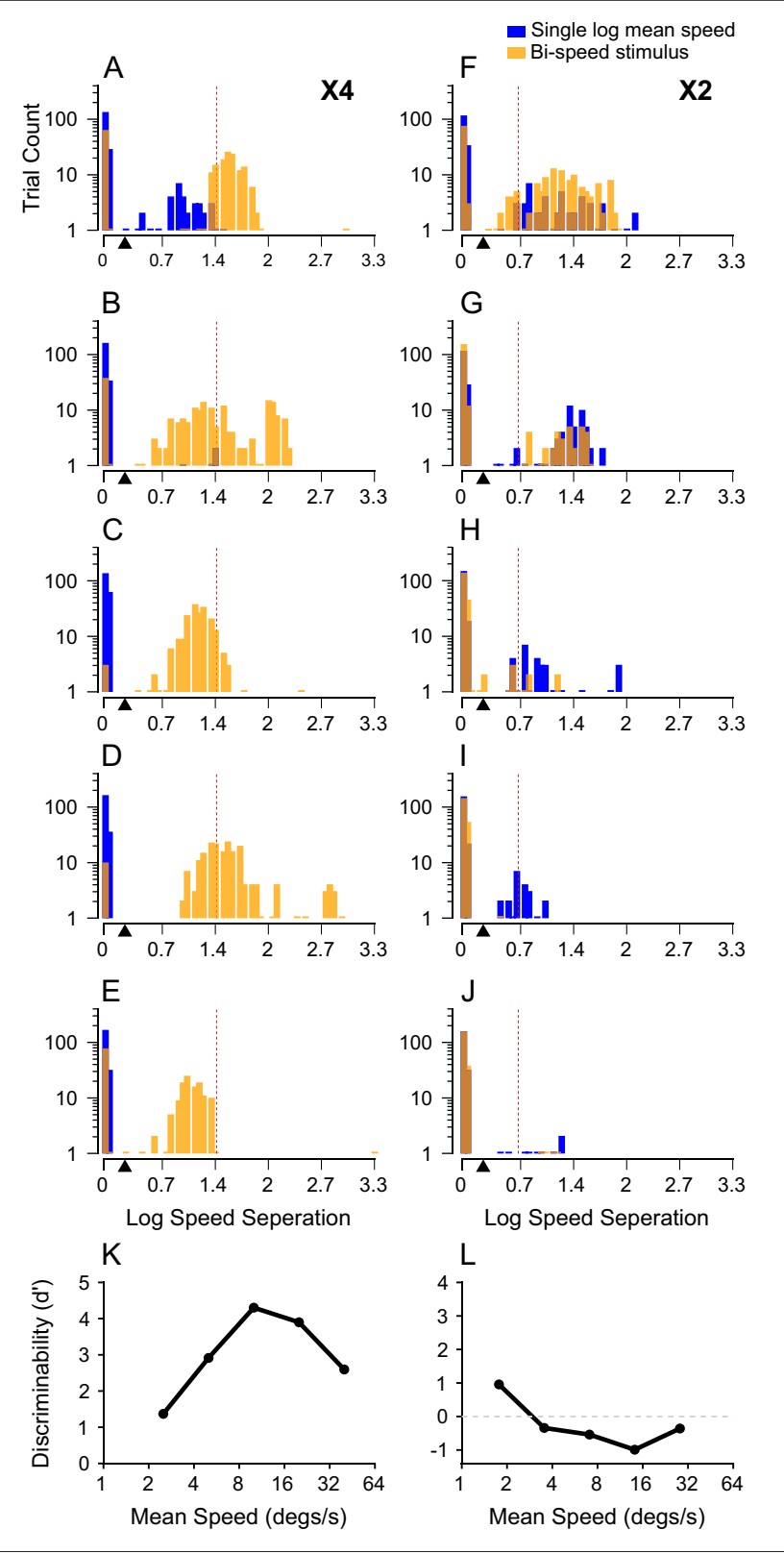

**Figure 12.** Discrimination between single- and bi-speed stimuli based on decoded speeds. (**A–J**) The distributions of the speed separation between two readout speeds in each trial in response to either the bi-speed stimuli (yellow) or the corresponding single, log-mean speed (blue). The abscissa is shown on a natural logarithm scale. The bin width is 0.05. The red dotted line indicates veridical speed separation. (**A–E**) ×4 speed separation. The

*Figure 12 continued on next page*

*Figure 12 continued*

speeds of the bi-speed stimuli are 1.25 and 5$^0$/s (**A**), 2.5 and 10$^0$/s (**B**), 5 and 20$^0$/s (**C**), 10 and 40$^0$/s (**D**), 20 and 80$^0$/s (**E**). (**F-J**) ×2 speed separation. The speeds of the bi-speed stimuli are 1.25 and 2.5$^0$/s (**F**), 2.5 and 5$^0$/s (**G**), 5 and 10$^0$/s (**H**), 10 and 20$^0$/s (**I**), 20 and 40$^0$/s (**J**). (**K–L**) The performance of discriminating a bi-speed stimulus from the corresponding log-mean speed is based on the speed separation of the decoded speeds. (**K**) ×4 speed separation; (**L**) ×2 speed separation. The black triangles in A-J indicate the speed separation threshold of 1.3x (0.26 on the log scale) used for discriminating bi-speed and single-speed stimuli.

corresponding single log-mean speed. As stated above, when the difference between the weights of two readout speeds in a trial was greater than 0.7, the trial was considered to have a single readout speed, and the speed separation was set to zero.

At 4x speed separation, the separations between the readout speeds extracted from the response to the bi-speed stimuli generally matched the veridical speed separation. They were larger than those extracted from the response to single log-mean speed (*Figure 12A–E*). Based on the distributions of the decoded speeds, we used a speed separation threshold of 1.3x (i.e. 0.26 on the log scale, marked by a black triangle in *Figure 12*) to distinguish single- and bi-speed stimuli and to evaluate the hit rate and false alarm rate. The choice of the threshold within a range from 1.1x to 1.7x did not change the results qualitatively. We calculated *d'* to measure the ability to discriminate the bi-speed stimuli from the corresponding single log-mean speed. The *d'* (*Figure 12K*) was similar to the psychophysical performance of the monkey subject (*Figure 2B1*), reaching its peak at 5 and 20°/s. Although *d'*s at stimulus speeds of 1.25 and 5°/s and 2.5 and 10°/s were smaller than those of human subjects (*Figure 1C1 and E*), the fact that in many trials, the readout speeds matched the faster stimulus speeds (*Figure 11A and B*) indicated that the decoder was able to segment the visual stimuli when stimulus speeds were low.

At 2x speed separation, except at 1.25 and 2.5°/s, the distribution of the speed separation extracted from the response to the bi-speed stimuli was similar to that extracted from the inferred response to single log-mean speed (see Methods) (i.e. orange and blue bars overlapping) (*Figure 12F–J*). The *d'* calculated based on the decoded speed separation (*Figure 12L*) was smaller than the psychophysical performance of human and monkey subjects (*Figure 1C2 and E2*; *Figure 2B2*), suggesting that the decoder was not able to segment the visual stimuli at 2x speed separation, with a potential exception at the lowest speeds of 1.5 and 2.5°/s.

## Discussion
### Perceptual segmentation of multiple motion speeds

Our human psychophysical study employed a novel 3AFC task. The task combined an identification task (to report whether a stimulus had one or two speeds) with a discrimination task (to compare a two-speed stimulus with a single-speed stimulus) (*Figure 1A, E and E2*). This approach allowed us to characterize discriminability based on perceptual segmentation, rather than other perceptual appearances of the stimuli. We made two findings. First and intuitively, the performance of speed segmentation was better when the separation between two stimulus speeds was larger. Second, at a fixed speed separation, speed segmentation became harder at fast speeds. Our results are consistent with previous studies. *Masson et al., 1999* showed that the speed segmentation threshold increased sharply when the mean stimulus speed increased from 8°/s to 16°/s. By varying the width of a speed notch, *Rocchi et al., 2018* showed that transparent motion perception was stronger with a wider notch width, and that transparent motion was well perceived at slow speeds (mean speed = 4.6°/s) but not at faster speeds (mean speed = 20.6°/s) at a range of notch widths from 1 to 6°/s. Our study tested a larger range of speeds and showed that the segmentation performance dropped sharply at speeds of 20 and 80°/s (4x), and 20 and 40°/s (2x), faster than those shown in the previous studies. This discrepancy is likely due to the larger speed separations used in our study and the difference in stimuli. The visual stimuli used in our study had either one or two speeds, whereas those used by *Rocchi et al., 2018* were sampled from a distribution of motion speeds and had multiple elements.

# Neural encoding of multiple speeds and implications for efficient coding

We found that, at low stimulus speeds, MT neurons showed a faster-speed bias in representing two speeds of overlapping stimuli. We also showed that faster-speed bias in MT is a robust phenomenon regardless of whether the stimulus components move in the same or different directions. A faster-speed bias in representing two motion speeds is a novel finding. It adds to a growing body of studies demonstrating that visual neurons do not necessarily average the responses elicited by individual stimulus components in response to multiple stimuli (e.g. *Ni et al., 2012*; *Bao and Tsao, 2018*). Our laboratory has previously reported that the responses of MT neurons to multiple moving stimuli can show a bias toward the stimulus component with a higher signal strength, such as motion coherence or luminance contrast (*Xiao et al., 2014*; *Wiesner et al., 2020*), a directional side bias toward one of two motion directions, even when the stimulus components have the same signal strength (*Xiao and Huang, 2015*), and a disparity bias toward one of two surfaces moving at different stereoscopic depths (*Chakrala et al., 2024*). These different response biases enhance the representation of individual stimulus components and can help to facilitate the segmentation of multiple moving stimuli (*Orhan and Ma, 2015*).

While the faster-speed bias reported in this study may facilitate the segregation of faster-moving stimuli, it may come at the cost of reduced ability to segregate slower speeds. Why does the primate visual system encode multiple speeds in this way? An efficient way to represent sensory information is to devote limited resources to better represent signals that occur more frequently in the natural environment (*Attneave, 1954*; *Barlow, 1961*; *Simoncelli and Olshausen, 2001*). Previous studies have suggested that slow speeds are more likely to occur than fast speeds in natural scenes (*Weiss et al., 2002*; *Stocker and Simoncelli, 2006*; *Zhang and Stocker, 2022*). If neurons in the primate visual cortex are optimized to efficiently represent speeds that are more likely to occur in natural scenes, one may expect to find neurons showing a slower-speed bias rather than a faster-speed bias. However, besides maximizing information about the environment, neural representation in the sensory cortices may be optimized for other goals, such as maximizing the performance of certain behavioral tasks (*Simoncelli and Olshausen, 2001*; *Manning et al., 2024*). Since a figural object tends to move faster than its background in natural scenes (*Huang et al., 2019*; *Friedman et al., 2025*), a neural representation of multiple motions with a faster-speed bias would help to identify the figure and, therefore, benefit the performance of an essential behavioral task – figure/ground segregation. Our finding of a faster-speed bias at slow stimulus speeds underscores the possibility that, when choosing between efficiently representing the most commonly occurring features in natural scenes (e.g. slow speeds) and enhancing behavioral performance in critical tasks (e.g. figure-ground segregation), some brain areas in the visual system may prioritize enhancing the behavioral performance.

# Potential mechanisms underlying the neural encoding of multiple speeds

We found that the faster-speed bias was still present when attention was directed away from the RFs, suggesting that the faster-speed bias cannot be explained by an attentional modulation. The faster-speed bias cannot be explained by the apparent contrast of the stimulus component either – the random dots of the faster-speed component had shorter dwell time on the video display and appeared dimmer than the slower component. We suggest a modified normalization model that may explain why the faster-speed bias in MT occurs at low stimulus speeds and diminishes at high speeds.

Previous studies that characterize the neural representation of multiple stimuli have used stimulus strength to weigh the component responses in the divisive normalization model (e.g. *Busse et al., 2009*; *Ni et al., 2012*; *Xiao et al., 2014*; *Heuer and Britten, 2002*). In comparison to the standard normalization model, we suggest that the response of a population of neurons (i.e. the weighting pool) defines the numerator of the normalization equation (*Equation 6*). The weighting pool may or may not be the same as the normalization pool that defines the response in the denominator. We suggest that the weighting pool contains a population of neurons with a broad range of speed preferences. In this way, the summed (or averaged) response of the weighting pool depends mainly on the stimulus speed, and therefore, the weighting is less sensitive to the individual neuron's speed preference. In this study, we used MT population-averaged responses to single speeds to approximate the responses of the weighting pool. MT population-averaged speed tuning in our data peaked

around 20°/s (*Figure 8E*), consistent with previous studies (*Maunsell and Van Essen, 1983*; *Lisberger and Movshon, 1999*; *Nover et al., 2005*; *Huang and Lisberger, 2009*). At stimulus speeds less than 20°/s, the population speed tuning has a positive slope, and a faster component would elicit a stronger population response than a slower component. This insight explains the faster-speed bias at low stimulus speeds and why it tends to be stronger at 4x than at 2x speed separation. Conceptually, this model can also explain why faster-speed bias diminishes at higher speeds. When two stimulus speeds are at opposite sides of the population's preferred speed, they elicit similar population responses in the weighting pool. This model also predicts that when both stimulus speeds exceed the preferred speed of the weighting pool, the response weight for the slower component should be greater than that for the faster component.

This modified normalization model well described our data on MT responses to two stimuli moving in different directions at 2.5 and 10°/s (*Figure 8*). However, our current data set has limitations to validate this model fully. This normalization model (*Equation 6*) can reasonably describe our data on MT responses to bi-speed stimuli moving in the same direction across five speed pairs (*Figure 5*) (results not shown). However, since the responses of each neuron to the bi-speed stimuli only have five data points (see *Figure 3*) and our model has four free parameters (*Equation 6*), the model is under-constrained. In future work, it will be important to extend the experiment to include pairs of stimuli moving in different directions at varying combinations of speeds across a broader range. By incorporating full direction tuning curves to better constrain the model (as shown in *Figure 8*) and systematically varying speed combinations, future studies could test the model's prediction that the response bias shifts from a faster-speed bias to response averaging and eventually to a slower-speed bias, as the stimulus speeds increase.

Although in our model, we used the responses of a population of MT neurons to estimate the responses of the weighting pool, the weighting pool may be composed of neurons that feed signals into MT and have similar population-averaged speed tuning as MT neurons. MT neurons receive feed-forward motion-selective input mainly from V1, and also from V2 and V3 (*Ungerleider and Desimone, 1986*; *Movshon and Newsome, 1996*; *Anderson et al., 1998*; *Anderson and Martin, 2002*; *Rockland, 2002*). Speed-selective complex cells in V1 have preferred speeds in a range similar to that of MT neurons, but the mean preferred speed is slower than MT (*Mikami et al., 1986*; *Orban et al., 1986*; *Priebe et al., 2006*). Future studies examining the transition speeds at which faster-speed bias changes to response averaging and slower-speed bias may help to differentiate whether the weighting pool consists of neurons in MT or early visual areas, such as V1.

## Decoding multiple speeds from population neural responses

Theoretical studies have proposed neural coding of probability distribution and multiplicity of a visual attribute (*Pouget and Snyder, 2000*). The key idea of this framework is that neurons do not code a single stimulus value but instead code the distribution of the stimulus (*Zemel et al., 1998*; *Pouget et al., 2003*). However, neurophysiological evidence supporting this framework on coding multiplicity is limited. Previous studies have not demonstrated the ability to extract multiple speeds from population neural responses. Our results provide support for this framework of coding multiplicity. Our decoding analysis reveals that the population neural response in MT carries information about multiple speeds of overlapping stimuli, and it is possible to extract multiple speeds and their weights even when the population neural response has a unimodal distribution.

At large (4x) speed separation, our decoding results captured several key features of humans' and monkeys' perception of multiple speeds – the decoded speeds support perceptual segmentation at low to intermediate speeds (*Figures 11A–E and 12A–K*). At 20 and 80°/s, the decoder was uncertain about whether a single speed or two speeds were present in the visual stimuli and, therefore, had difficulty segmenting the visual stimuli at these fast speeds (*Appendix 1—figure 4*). However, at small (2x) speed separation, the decoding results showed very little segmentation (*Figures 11G–J and 12L*), except at very low speeds. This result differs from the perception at stimulus speeds less than 20 and 40°/s (*Figures 1C2, E2 and 2B2*). What are the potential reasons for the decoder's inadequacy in segmenting small speed separations? The best-fit population response predicted by the encoding rule of the decoder matched the neural responses remarkably well ($R^2 > 0.99$ for all five speed pairs of 2x separation, *Figure 10F–J*). So, the encoding model for decoding effectively described the population neural responses to the bi-speed stimuli. Because we found the same results when performing

decoding based on neural responses averaged across experimental trials (*Figure 10G–J*), this inadequacy was unlikely due to our assumption of the trial-by-trial response variability following a Poisson process. We consider several factors that may contribute to this discrepancy.

First, this may be attributed to the limited sample size of our dataset. If we had a much larger MT neuron population, potential differences in neuronal responses to bi-speed stimuli and the single log mean speed might be captured by the data, which may lead to better decoding. Second, it may be due to the choice of the 'objective function.' Our decoder minimized the sum of squared error between the predicted population response and the recorded neural response. In contrast, *Zemel et al., 1998* found motion directions that maximized the posterior probability $P(s|r)$ using a maximum a posteriori (MAP) estimate. It remains to be determined whether maximizing the posterior probability can improve the resolution of segmenting multiple speeds. Third, in different sensory areas, neuronal responses to two stimuli can fluctuate from trial to trial, representing one stimulus component over the other (*Li et al., 2016*; *Caruso et al., 2018*; *Jun et al., 2022*; *Schmehl et al., 2024*; *Groh et al., 2024*). If this trial-varying stimulus multiplexing also occurred for representing two speeds with a small separation, information about individual speed components would be lost in the trial-averaged responses (with added variability based on a Poisson process), as in our decoding procedure. Future studies with a large number of repeated experimental trials would be needed to test this possibility. Finally, while area MT is clearly important for motion-based segmentation, other motion-sensitive brain areas may be important for segmenting speeds with a small difference.

## Materials and methods

We conducted psychophysical experiments using human subjects, and psychophysical and neurophysiological experiments using macaque monkeys.

### Human psychophysics

#### Subjects
Four adult human subjects (*CN, CO, IN, NP*), two men and two women, with normal or corrected-to-normal visual acuity, participated in the psychophysics experiments. Subject *CN* was naive about the purposes of the experiments. Subjects CO and IN had a general idea about this study but did not know the specific design of the experiments. Informed consent was obtained from the subjects. All aspects of the study were in accordance with the principles of the Declaration of Helsinki and were approved by the Institutional Review Board at the University of Wisconsin-Madison (2015–0808).

#### Apparatus
Visual stimuli were generated by a Linux workstation using an OpenGL application and displayed on a 19-inch CRT monitor. The monitor had a resolution of 1024×768 pixels and a refresh rate of 100 Hz. The output of the video monitor was measured with a photometer (LS-110, Minolta) and was gamma-corrected. Stimulus presentation was controlled by a real-time data acquisition and stimulus control program 'Maestro' (https://sites.google.com/a/srscicomp.com/maestro/) as in the animal behavior and neurophysiology experiments. Subjects viewed the visual stimuli in a dark room with dim background illumination. The viewing distance was 58 cm. A chin rest and forehead support were used to restrict the head movements of the observers. During experimental trials, human subjects maintained fixation on a small spot within a 2 × 2° window. Eye positions were monitored using a video-based eye tracker (EyeLink, SR Research) at a rate of 1 kHz.

#### Visual stimuli
Visual stimuli were two spatially overlapping random-dot patches presented within a square aperture 10° wide. Each square stimulus was centered 11° to the right of the fixation spot, therefore, covering 6° to 16° eccentricity. This range roughly matched the RF eccentricity of the recorded MT neurons in our neurophysiological experiments. The random dots were achromatic. Each random dot was 3 pixels and had a luminance of 15.0 cd/m². The background luminance was 0.03 cd/m². The dot density of each random dot patch was 2 dots/degree². The two random-dot patches translated horizontally in the same direction. To reduce adaptation, the motion direction was either leftward or rightward in half of the trials, and stimulus trials were randomly interleaved. In one set of trials, two overlapping

random-dot patches had a 'large speed separation' and the speed of the faster component was always four times (4x) that of the slower component. In another set of trials, visual stimuli had a 'small speed separation' and the speed of the faster component was always twice (2x) that of the slower component (see *Figure 1B1 and B2*). For each bi-speed stimulus, there was a corresponding single-speed stimulus composed of two overlapping random-dot patches moving in the same direction at the same speed. The single speed was the natural logarithmic (log) mean speed of the bi-speed stimulus: $Spd_{mean} = e^{[ln(Spd_1)+ln(Spd_2)]/2}$, in which $Spd_1$ and $Spd_2$ were the two component speeds. The motion coherence of each random-dot patch was always 100%.

## Procedure

In a standard two-alternative-forced-choice (2AFC) task, subjects discriminated a bi-speed stimulus from the corresponding single log-mean speed stimulus. The bi-speed and single-speed stimuli were presented in two consecutive time intervals with a 500 ms gap in between, in random, balanced order. In each time interval, the visual stimulus appeared, remained stationary for 250 ms, and then moved for 500 ms. At the end of each trial, subjects reported which time interval contained a bi-speed stimulus by pressing one of two buttons (left or right) within a 1500 ms window. After the button press, the inter-trial interval was 1300 ms. Each block of trials contained 40 trials, i.e., 5 speed pairs × 2 speed separations × 2 temporal orders (the bi-speed stimulus appeared in the first or second time interval) × 2 motion directions (visual stimuli moved either to the left or right). Each experimental session typically contained five blocks, i.e., 200 trials.

Subjects also performed a 3AFC task. As in the 2AFC task, subjects discriminated a bi-speed stimulus from the corresponding single log-mean speed stimulus but had the option to make a third choice by pressing the middle button on trials when they thought neither stimulus interval appeared to contain two speeds ('no two-speeds' choice). When subjects thought one of the two stimulus intervals contained two speeds, subjects then pressed either the left or the right button to indicate which interval had two speeds.

## Data analysis

The hit rate was calculated as the percentage of trials in which a subject correctly picked the bi-speed stimulus as having two speeds. The false alarm rate was calculated as the percentage of trials in which a subject incorrectly identified the single-speed stimulus as having two speeds. As a measure of discriminability between the bi-speed and the corresponding single-speed stimuli, we calculated the discriminability index $d'=norminv$(hit rate) – $norminv$(false alarm rate). $norminv$ is a MATLAB function that calculates the inverse of the normal cumulative distribution function, with the mean and standard deviation set to 0 and 1, respectively. When the hit or false alarm rate was occasionally close to 1, to avoid infinite d' values, d' was calculated using a modified formula: $d'=norminv\{[(100 \times$ hit rate)+1]/102\} - norminv\{[(100 \times$ false alarm rate)+1]/102\}. In analyzing the results of the 3AFC task, we incorporated the NTC trials into the $d'$ calculation by evenly splitting the NTC trials into 'hit' trials and 'false alarm' trials. In this way, the NTC trials were still accounted for by the hit rate and false alarm rate, in the sense that they did not contribute to the discrimination. We also examined the percentage of trials in which subjects made the NTC choice at different stimulus speeds.

## Neurophysiological and psychophysical experiments

### Subjects

Five male adult rhesus monkeys (*Macaca mulatta*) were used in the experiments. Four monkeys were used in the neurophysiological experiments, and one was used in the psychophysical experiment. Experimental protocols were approved by the Institutional Animal Care and Use Committee at the University of Wisconsin–Madison (G005924), and were in strict compliance with U.S. Department of Agriculture regulations and the National Institutes of Health *Guide for the Care and Use of Laboratory Animals*.

### Apparatus and electrophysiological recording

Procedures for surgical preparation and electrophysiological recording were routine and similar to those described previously (*Huang and Lisberger, 2009*; *Xiao et al., 2014*). For subjects IM and MO,

horizontal and vertical eye positions were monitored using the search coil method at a sampling rate of 1 kHz on each channel. For subjects RG, GE, and BJ, eye positions were monitored using a video-based eye tracker (EyeLink, SR Research) at a rate of 1 kHz. For electrophysiological recordings, we lowered single-contact tungsten microelectrodes (Thomas Recording or FHC) either using the Mini-Matrix microdrive (Thomas Recording) or the NAN drive (NAN Instruments) into the posterior bank of the superior temporal sulcus. The impedances of the electrodes were 1~3 MΩ. We identified area MT by its characteristically large proportion of directionally selective neurons, small classical RFs relative to those in the neighboring medial superior temporal area, and location on the posterior bank of the superior temporal sulcus. Electrical signals were filtered, amplified, and digitized conventionally. Single units were identified with a real-time template-matching system (Plexon). Spikes were carefully sorted using the Plexon offline sorter.

Stimulus presentation and the behavioral paradigm were controlled by a real-time data acquisition program Maestro as described in the human psychophysics experiment. For neurophysiological recordings from IM and MO, visual stimuli were presented on a 20-inch CRT monitor at a viewing distance of 38 cm. Monitor resolution was 1280×1024 pixels and the refresh rate was 85 Hz. For RG, GE, and BJ, visual stimuli were presented on a 25-inch CRT monitor at a viewing distance of 63 cm. Monitor resolution was 1024×768 pixels and the refresh rate was 100 Hz. Visual stimuli were generated by a Linux workstation using an OpenGL application that communicated with the main experimental-control computer over a dedicated Ethernet link. The output of the video monitor was gamma-corrected.

## Visual stimuli and experimental procedure of the main experiment

All visual stimuli were presented in individual trials while monkeys maintained fixation. Monkeys were required to maintain fixation within a 1.5×1.5° window centered around a fixation spot during each trial to receive juice rewards, although actual fixation was typically more accurate. In a trial, visual stimuli were illuminated after the animal had acquired fixation for 200 ms. To assist the isolation of directional-selective neurons in area MT, we used circular translation of a large random-dot patch (30×30°) as a search stimulus (*Schoppmann and Hoffmann, 1976*). After an MT neuron was isolated, we characterized the direction tuning using randomly interleaved trials of 30×30° random-dot patches moving at 10°/s in eight different directions, ranging from 0 to 315° in 45° steps. Next, we mapped the RF by recording responses to a series of 5×5° patches of random dots that moved in the preferred direction of the neuron at 10°/s. The location of the patch was varied randomly to tile the screen in 5° steps without overlap, covering an area of either 40×30° or 35×25°. The raw map of the RF was interpolated using the Matlab function *interp2* at an interval of 0.5°, and the location giving rise to the highest firing rate was taken as the center of the RF. In the following experiments, testing stimuli were centered on the RF.

Monkeys IM and MO were tested with the main visual stimuli used in our experiments, which were two spatially overlapping random-dot patches presented within a square aperture 10° wide. The random dots were achromatic. The dot density of each random-dot patch was 2 dots/degree². Each random dot was 3 pixels at a side and had a luminance of 15.0 cd/m². The background luminance was <0.2 cd/m². In each trial, the random dots moved within the aperture. The two random-dot patches were translated at two different speeds, both at 100% motion coherence, in the same direction (the preferred direction of the recorded neuron). The ratio between the two component speeds was fixed either at 4 (i.e. the large speed separation) or 2 (i.e. the small speed separation) (see Methods for human psychophysics above). At 4x speed separation, the five speed pairs used were 1.25 and 5°/s, 2.5 and 10°/s, 5 and 20°/s, 10 and 40°/s, and 20 and 80°/s (*Figure 1B1*). At 2x speed separation, the speed pairs used were 1.25 and 2.5°/s, 2.5 and 5°/s, 5 and 10°/s, 10 and 20°/s, and 20 and 40°/s (*Figure 1B2*). Experimental trials of bi-speed stimuli that had 4x or 2x speed separations were randomly interleaved. Also randomly interleaved were trials that showed only a single random-dot patch moving at a speed of 1.25, 2.5, 5, 10, 20, 40, or 80°/s, which were the individual stimulus components of the bi-speed stimuli.

Monkeys RG and GE were tested with a variation of the main visual stimuli, where two overlapping random-dot stimulus components moved at fixed speeds of 2.5 and 10°/s, respectively, in two different directions separated by 90°. The diameter of the stimulus aperture was 3°. The faster component moved at the clockwise side of the two component directions (illustrated in *Figure 8*). We varied

the vector average direction of the two stimulus components across 360° in a step of 15° to characterize the direction-tuning curves of MT neurons. We also measured the direction-tuning curves to a single stimulus moving at the individual component speeds.

## Behavioral paradigm and visual stimuli of attention control

Monkey RG was also tested in a control experiment in which the attention of the animal was directed away from the RFs of MT neurons. The attended stimulus was a random-dot patch moving in a single direction at 100% motion coherence within a stationary circular aperture that had a diameter of 5°. The stimulus patch was centered 10° to the left of the fixation spot, in the visual hemifield contralateral to the hemifield of the recorded MT neurons' RFs. The monkey performed a direction discrimination task to report whether the motion direction of the attended stimulus moved at the clockwise or counter-clockwise side relative to the vertical direction. While the animal fixated on a point at the center of the monitor, both the attended stimulus and the RF stimulus were turned on and remained stationary for 250 ms before they moved for 500 ms. The attended stimulus translated at a speed of 10°/s and in a direction either clockwise or counter-clockwise from an invisible vertical (upward) direction by an offset of 10°, 15°, or 20°. The RF stimuli were the same as our main visual stimuli, with either a single-speed or bi-speed stimulus moving in the same direction. All trials were randomly interleaved. After the motion period, all the visual stimuli were turned off, and two reporting targets appeared 10° eccentric on the left and right sides of the fixation point. To receive a juice reward, the animal was required to make a saccadic eye movement within 400 ms after the fixation spot was turned off, either to the left or right target, depending on whether the motion direction of the attended stimulus was counter-clockwise or clockwise to the vertical direction, respectively.

## Monkey psychophysics

Monkey BJ was trained to perform a 2AFC discrimination task. The visual stimuli were the same as our main visual stimuli in the neurophysiological experiments, except that the stimulus moving at a single speed was also composed of two overlapping random-dot patches moving in the same direction at the same speed, the same as in the human psychophysics experiments. In this way, the single-speed stimulus and the bi-speed stimuli had the same dot density. Visual stimuli were random-dot patches moving within a square aperture of 10°×10°, centered 10° to the right of the fixation spot. The motion direction of the visual stimuli was always rightward. Experimental trials of bi-speed stimuli that had 4x or 2x speed separations, as well as the single-speed stimulus that moved at the log mean speed of the bi-speed stimuli, were randomly interleaved. Visual stimuli were turned on, remained stationary for 250 ms, and then moved for 500 ms. Following the stimulus offset, two reporting targets (dots) were presented 5.7° away from the fixation spot, at upper right (4°, 4°) and lower left (–4°, –4°) positions relative to the fixation spot. To receive a juice reward, the animal was required to make a saccadic eye movement to one of the two targets within 300 ms after the fixation spot was turned off. In most of the experiment trials, the animal received juice rewards for selecting the upper-right target when visual stimuli moved at two different speeds, and for selecting the lower-left target when visual stimuli moved at a single speed. Guided by our human psychophysics results, we made an exception to always reward the animal when the bi-speed stimuli moved at 20 and 80°/s or at 20 and 40°/s, regardless of which target was selected to avoid biasing the monkey's choice by veridically rewarding the animal. This was because, at these fast speeds, human subjects could not segment the bi-speed stimuli. During training, the animal was never presented with the bi-speed stimuli of 20 and 80°/s, and 20 and 40°/s. During testing, the trials of 20 and 80°/s, and 20 and 40°/s were randomly interleaved with bi-speed and single-speed trials that were rewarded veridically to anchor the task rule. Among all testing trials, only 10% of the trials were rewarded with a 100% rate. We collected 50 trials of data for 4x speed separation across five experimental sessions, and 90 trials for 2x speed separation across nine sessions during the testing phase. The hit rate, false alarm rate, and *d'* were calculated in the same way as in the human psychophysics experiments.

## Model fit of the tuning curves to bi-speed stimuli

We used a linear weighted summation model (*Equation 5*) to fit the direction-tuning curves to overlapping stimuli moving in different directions and at different speeds. We also fitted the direction-tuning curves to the bi-speed/bi-directional stimuli using a modified divisive normalization model (*Equation*

6). These model fits were obtained using the constrained minimization tool 'fmincon' (MATLAB) to minimize the sum of squared error. To evaluate the goodness of fit of models for the response tuning curves, we calculated the percentage of variance (PV) accounted for by the model as follows: PV = $100 \times (1 - \frac{SSE}{SST})$, where SSE is the sum of squared errors between the model fit and the neuronal data, and SST is the sum of squared differences between the data and the mean of the data.

## Construction of population neural response

For each recorded MT neuron, we plotted the trial-averaged speed tuning curve in response to the single speed and spline-fitted the tuning curve using the Matlab function *csaps* with the smoothing parameter *p* set to 0.93. We found *p*=0.93 best captured the trend of the speed tuning, without obvious overfitting. We then found the preferred speed of the neuron, defined as the speed at which the maximum firing rate was reached in the spline-fitted tuning curve. The neuron's responses to all single-speed and bi-speed stimuli were normalized by the maximum firing rate at the preferred speed. To construct the population neural response to a given stimulus, we took the normalized firing rate of each neuron elicited by that stimulus and plotted it against the preferred speed of the neuron. Because the preferred speeds of the neurons in our data sample did not cover the full speed range evenly, we spline-fitted (with a smoothing parameter of 0.93) the population neural response to capture it evenly across the full range of preferred speed.

## Discrimination of population neural responses using a classifier

We trained a linear classifier to discriminate between the constructed population neural responses to a bi-speed stimulus and those to the corresponding single-speed stimulus moving at the log mean speed. Constructed trial-by-trial population responses were generated randomly according to a Poisson process with the mean set to the recorded neuronal response averaged across experimental trials. For each speed combination, we generated 200 trials of responses to both the bi-speed stimulus and the corresponding single-speed stimulus. Constructed population responses were partitioned into training and testing sets using k-fold cross-validation (k=40). The 200 generated trials were randomly divided into 40 folds. The classifier was trained on 39 data folds and tested on the remaining fold, and the process was repeated 40 times to ensure that each fold was used for testing exactly once. The MATLAB *fitclinear* function was used to fit a linear classifier to the training data. The logistic learner and lasso regularization techniques were specified during the model training. The Stochastic Gradient Descent solver was used to optimize the objective function during the training of the classifier. The performance of the classifier was evaluated by *d'*, calculated using the hit rate and false alarm rate as described in human psychophysics.

## Population decoding

We define a given probability distribution of stimulus speed as: $\varnothing_m = \{P_{m,j}\}$, in which $P_{m,j}$ is the probability of speed $S_j$, j=1, 2, 3, ..., 121, and j evenly samples speeds from 1.25°/s to 80°/s (referred to as the 'full speed range') on a natural logarithm scale and at a '*speed interval*' of 0.0347. Because $\varnothing_m$ is a probability distribution, $\sum_j P_{m,j} = 1$. *m* is an index for different distributions.

The estimated response (*ES*) of neuron *i* to the stimulus speeds with a probability distribution $\varnothing_m$ is a linear sum of the responses of neuron *i* to each single speed $S_j$ within the full speed range, weighted by the probability of each speed in $\varnothing_m$. The probability can also be considered as the weight (signal strength) of the speed.

$$ES_i(\varnothing_m) = \sum_j P_{m,j} f_i(S_j), \tag{7}$$

where $f_i$ is the spline-fitted speed tuning curve of neuron *i* in response to single speeds.

The estimated population response (*EP*) of *N* neurons to $\varnothing_m$ is:

$$EP_m(\ln(PS_i)) = ES_i(\varnothing_m), \tag{8}$$

where $PS_i$ is the preferred speed of neuron *i*, $i = 1, 2, 3, ..., N$. N=100 in our neural data.

We then spline-fitted the estimated population response $EP_m(\ln(PS_i))$ using a smoothing parameter of 0.93, interpolating the PS within the full speed range from 1.25°/s to 80°/s in natural logarithm

with 121 evenly spaced values. The spline-fitted estimated population response is represented as $spEP_m\left(\ln\left(PS_j\right)\right)$, $i = 1, 2, 3, \ldots, 121$.

Similarly, we spline-fitted the recorded and normalized population neural response $RP_m\left(\ln\left(PS_i\right)\right)$, $i = 1, 2, 3, \ldots, 100$, and interpolated the PS to the same 121-speed values on a logarithm scale within the full speed range as above. The spline-fitted, recorded population neural response is represented as $spRP_m\left(\ln\left(PS_j\right)\right)$, $.j = 1, 2, 3 \ldots, 121$.

The decoded probability distribution of the stimulus speed $\varnothing_e$ is the $\varnothing_m$ that maximizes the objective function (OF), which is defined as the negative value of the SSE (sum squared error) between the spline-fitted estimated population response and the recorded neural response:

$$OF(\varnothing_m) = - \sum_j \left\{ \left[ spEP_m\left(\ln(PS_j)\right) - spRP_m\left(\ln(PS_j)\right)\right]^2 \right\}, \tag{9}$$

$$\varnothing_e = \arg\max{}_{\varnothing_m}[OF(\varnothing_m)]. \tag{10}$$

Rather than finding an arbitrary distribution, we constrained $\varnothing_e$ to contain either a single speed with a probability (referred to as the 'weight') of 1 or two speeds with the same or different weights that sum to 1.

## An algorithm to search for the probability distribution of stimulus speed

We first searched for the best-fit distribution $\varnothing_{e1}$ that contained a single speed $SP$ with non-zero probability ($p=1$) and gave rise to the maximum OF across the full speed range ($OF_{max1}$). We next searched for the best-fit distribution $\varnothing_{e2}$ that contained two speeds $SP_1$ and $SP_2$ with non-zero probability and gave rise to the maximum OF for two speeds ($OF_{max2}$). We varied the speed separation, the center position (i.e. the log mean speed), and the probabilities of the two speeds. For each speed separation and center position, the probabilities of $SP_1$ and $SP_2$ were varied from 0 to 1 at a step of 0.01, with the constraint that they summed to 1. We searched through the speed separation, $\ln(SP_2)-\ln(SP_1)$ from 0.0693 (i.e. 2 *speed intervals*) to 3.3271 (i.e. 96 *speed intervals*), in a step of 0.0693. The search range covered the speed ratio $SP_2/SP_1$ from 1.07x to 27.86x, sufficiently broader than 2x and 4x used in our visual stimuli. For each speed separation, we started the search where the center position of the two speeds $[\ln(SP_1)+\ln(SP_2)]/2$ was in the middle of the 121 possible speed values, referred to as the 'speed axis.' We then moved the center position toward the left border of $\ln(1.25)$ at a step of 0.0347 to find the maximum OF value ($OF_{leftmax}$) along the left half of the speed axis. If the OF value at the center position next to the current position was higher, the search moved to the next position. Otherwise, the current position was considered a local maximum. After we found a local maximum, the search continued in the same direction for up to another 30 *speed intervals*. This continued until one of the component speeds hit a border, 30 intervals were reached, or an OF value greater than the previous local maximum was found. If a larger OF was found, the local maximum was updated and the search jumped to that position, and the procedure was repeated until $OF_{leftmax}$ was found. We then returned to the middle of the speed axis and searched through speed pairs toward the right border $\ln(80)$ to find the maximum $OF_{rightmax}$. The larger one of $OF_{leftmax}$ and $OF_{rightmax}$ was the maximum OF for two speeds ($OF_{max2}$). The $\varnothing_e$ was either $\varnothing_{e1}$ or $\varnothing_{e2}$, whichever gave rise to the larger value of $OF_{max1}$ and $OF_{max2}$.

## Acknowledgements

We thank Dr. Steven Lisberger for his support in the early phase of this project, Emily Ausloos and Jianbo Xiao for data collection in early human psychophysics experiments, Bryce Arseneau for animal training, and Ying Cao for collecting additional neural data. We also thank Drs. Jennifer Coonen and Kevin Brunner at the Wisconsin National Primate Research Center for excellent veterinary care and surgical assistance, Kechen Zhang for helpful suggestions on the study, and Emily Cooper and Greg DeAngelis for their valuable comments on the manuscript. Research reported in this publication was supported by the National Eye Institute of the NIH grant R01EY022443 (XH). The content is solely the responsibility of the authors and does not necessarily represent the official views of the NIH.

# Additional information

## Funding

| Funder | Grant reference number | Author |
|---|---|---|
| National Eye Institute | R01EY022443 | Xin Huang |

The funders had no role in study design, data collection and interpretation, or the decision to submit the work for publication.

## Author contributions

Xin Huang, Conceptualization, Data curation, Formal analysis, Supervision, Funding acquisition, Investigation, Visualization, Methodology, Writing – original draft, Project administration, Writing – review and editing; Bikalpa Ghimire, Steven Wiesner, Data curation, Formal analysis, Investigation, Visualization, Methodology, Writing – original draft, Writing – review and editing; Anjani Sreeprada Chakrala, Data curation, Formal analysis, Investigation, Visualization, Methodology, Writing – original draft

## Author ORCIDs

Xin Huang https://orcid.org/0000-0001-9788-5383
Bikalpa Ghimire https://orcid.org/0000-0002-9004-3201
Anjani Sreeprada Chakrala https://orcid.org/0000-0002-7579-8139
Steven Wiesner https://orcid.org/0000-0002-8528-1096

## Ethics

Informed consent was obtained from the subjects. All aspects of the study were in accordance with the principles of the Declaration of Helsinki and were approved by the Institutional Review Board at the University of Wisconsin-Madison (2015-0808).

Experimental protocols were approved by the Institutional Animal Care and Use Committee at the University of Wisconsin-Madison (G005924) and were in strict compliance with U.S. Department of Agriculture regulations and the National Institutes of Health Guide for the Care and Use of Laboratory Animals.

Reviewer #1 (Public review): https://doi.org/10.7554/eLife.94835.4.sa1
Reviewer #3 (Public review): https://doi.org/10.7554/eLife.94835.4.sa2
Author response https://doi.org/10.7554/eLife.94835.4.sa3

# Additional files

## Supplementary files

MDAR checklist

## Data availability

The datasets and neural data files for this study, along with the MATLAB code used for data analysis, are available on OSF (https://osf.io/dg5n9).

The following dataset was generated:

| Author(s) | Year | Dataset title | Dataset URL | Database and Identifier |
|---|---|---|---|---|
| Wiesner S, Ghimire B | 2025 | Neural coding of multiple motion speeds in visual cortical area MT | https://osf.io/dg5n9 | Open Science Framework, dg5n9 |

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

# Appendix 1

## Additional analysis of the weight for the fast component, using the firing rates in response to the slower component from split trials

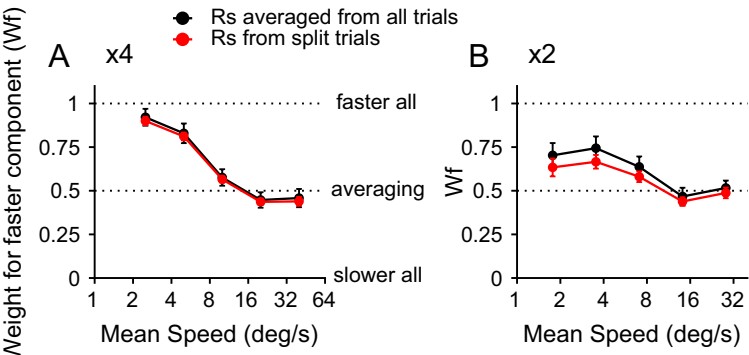

**Appendix 1—figure 1.** Weights for the faster stimulus component obtained from the slope of the linear regression, based on the recorded responses of 100 neurons. The black curves are a replot of those in *Figure 5F1 and F2*, with the $Rs$ (the response to the slower component) calculated based on the average firing rate across all trials for each neuron at each speed pair. The red curves were calculated the same way as the black curves, except that the $Rs$ used for the linear regression between $(R - R_s)$ and $(R_f - R_s)$ was averaged across different subsets of trials. For each neuron and at each speed pair, we determined $R_{s1}$ by averaging the firing rates of $R_s$ across half of the recorded trials, selected randomly. If the number of trials (**n**) was odd, we selected (n+1)/2 trials. We also determined $R_{s2}$ by averaging the firing rates of $R_s$ across the rest of the trials. We regressed $(R - R_{s1})$ on $(R_f - R_{s2})$, as well as $(R - R_{s2})$ on $(R_f - R_{s1})$, and repeated the procedure 50 times. We then averaged the slope for each speed pair across a total of 100 regressions. Red error bars represent the standard deviation (N=100 regressions). (**A**) 4x speed separation. (**B**) 2x speed separation.

## Behavioral performance of the fine-direction discrimination task and MT response properties when the attention was directed away from MT neurons' RFs

The monkey RG performed a direction discrimination task with an average correct rate of 86.7 ± 7.3% (mean ± std) across 23 sessions and over 5000 trials. The correct rates for 10°, 15°, and 20° direction offsets of the direction discrimination task were 78.8 ± 9.7%, 87.5 ± 8.3%, and 93.9 ± 5.8%, respectively (see Methods).

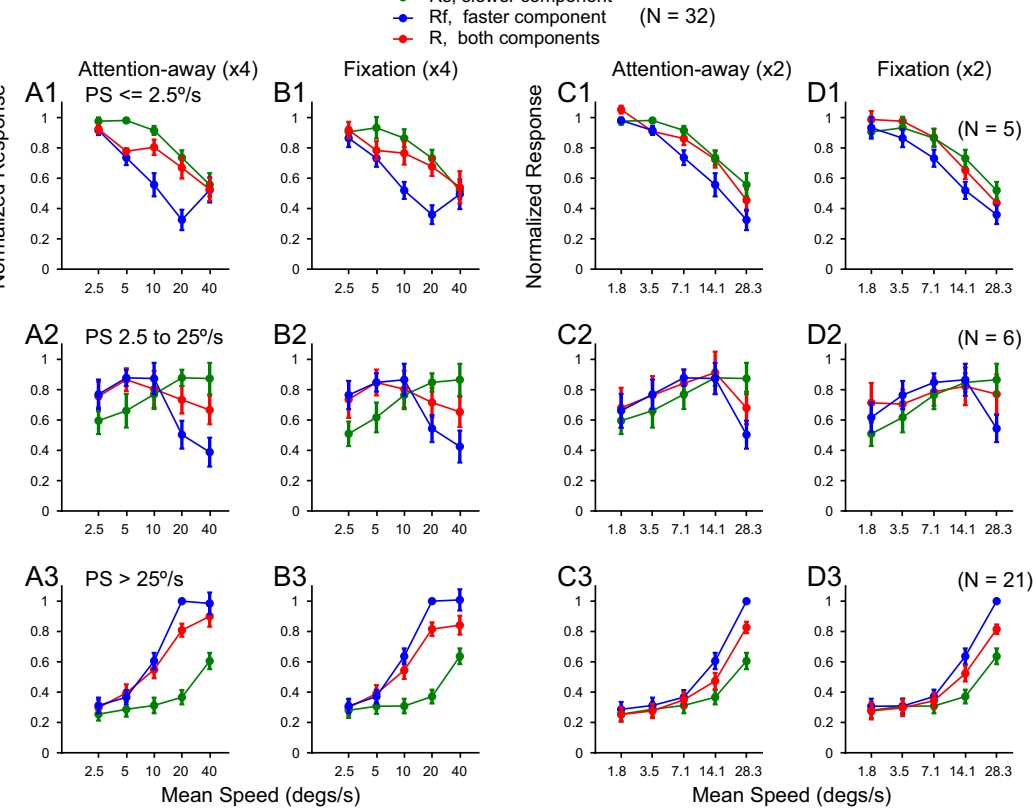

**Appendix 1—figure 2.** Population-averaged speed tuning curves to bi-speed stimuli and constituent single-speed components recorded in an attention-away and a fixation paradigm. Speed tuning curves from one monkey (RG) averaged across: (**A1–D1**) five neurons that had preferred speed (PS) ≤2.5°/s, (**A2–D2**) six neurons that had PS between 2.5 and 25°/s, (**A3–D3**) 21 neurons that had PS >25°/s. Error bars represent ± STE. (**A1–A3**) and (**B1–B3**) 4x speed separation; (**C1–C3**) and (**D1–D3**) 2x speed separation. (**A1–A3**) and (**C1–C3**) Attention directed away from the receptive fields (RFs); (**B1–B3**) and (**D1–D3**) Fixation paradigm.

# Trial-by-trial readouts from population neural responses to single speeds

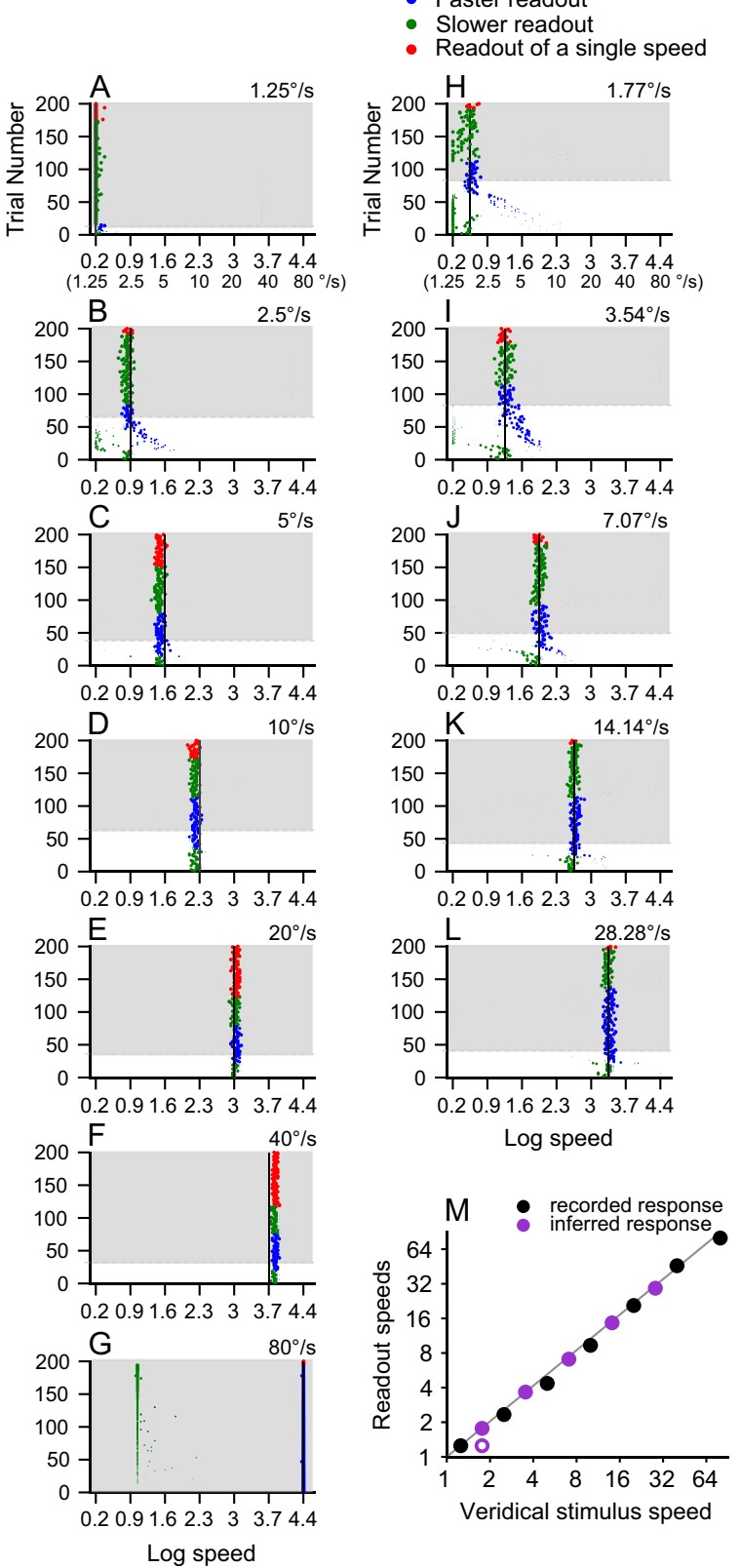

*Appendix 1—figure 3 continued on next page*

**Appendix 1—figure 3.** Trial-by-trial readout speeds decoded from population neural responses to single speeds. The neural population contained 100 recorded neurons, as shown in *Figure 9*. The trial-by-trial responses were randomly generated based on a Poisson process, with the mean set to the spike count averaged across the recorded trials. Each row in A-L shows the readout speed(s) from one trial, and each dot's size is proportional to the weight of the readout speed. If only one speed is decoded in a trial, that readout speed is shown in red. In trials with two readout speeds, the slower and faster readout speeds are shown in green and blue, respectively. The white background indicates trials with a weight difference between two readout speeds of less than 0.7, which are considered to have two readout speeds. The gray background indicates trials with a weight difference greater than 0.7, which are considered to have only one readout speed. The vertical black line and the speed marked in each panel indicate the stimulus speed. (**A–G**) Speeds decoded from recorded population neural responses to single speeds from 1.25 to 80°/s. Note that, at the stimulus speed of 80°/s (**G**), in addition to picking up the veridical speed of 80°/s (log speed of 4.382), the decoder often picked up a slower speed at 2.872°/s (log speed of 1.055), which was at the largest speed separation from 80°/s used in our searching algorithm (27.86x, log value 3.327). This border effect can also be seen at the stimulus speed of 1.25°/s (**A**), where a weaker and faster speed was sometimes picked up around 34.8°/s (log speed of 3.55). (**H–L**) Speeds decoded from inferred population neural response to single speeds, which are the log-mean speeds of the bi-speed stimuli with 2x speed separation. The responses of these log mean speeds of 2x speed separation were obtained from the splined-fitted, trial-averaged speed-tuning curve of each neuron. (**M**) Comparison of the readout speeds and the stimulus speeds. The diagonal line is the unity line. The ordinate represents the speed at the peak of the readout speed distribution pooled across simulated trials (not shown). At the stimulus speed of 1.77°/s (**H**), the distribution of the readout speed has two peaks, indicated by a solid circle (at 1.77°/s) and an open circle (at 1.25°/s). At the stimulus speed of 80°/s (**G**), the distribution of the readout speed also has two peaks; only the readout speed for the higher peak is shown in M.

## Analysis of decoded speeds of the bi-speed stimulus with the speeds of 20 and 80°/s

At the fastest stimulus speeds of 20 and 80°/s, across all trials, the mean objective function value peaked at speed separation of 3.25x (mean OF = –0.17, std = 0.14) (purple vertical line in *Appendix 1—figure 4A*). However, the peak value is not significantly different from the mean objective function value at the largest speed separation (27.86x, 3.3 on the log scale) searched (mean OF = –0.19, std=0.14) (paired t-test, *p*=0.31) (orange vertical line in *Appendix 1—figure 4A*). The flat objective function suggests high uncertainty of the extracted speed separation at this speed pair.

We divided the trials into two subgroups, based on whether they had one or two readout speeds, and calculated the objective function for each subgroup.

For trials considered to have two readout speeds, the objective function peaked at the speed separation of 3.25x (1.18 on the log scale) (purple vertical line in *Appendix 1—figure 4B*), corresponding to two readout speeds of 17.8 and 58.0°/s (the intersection points between the purple vertical line with the thick navy blue and thick green curves) (*Appendix 1—figure 4C*).

For trials considered to have one readout speed, the mean objective function showed a peak at the speed separation of 27.86x (3.3 on the log scale), which was the largest speed separation searched (orange vertical line in *Appendix 1—figure 4E*). As shown in *Appendix 1—figure 4F*, the dominant faster readout speed approached the log mean speed (40°/s, 3.7 on log scale) (thick navy blue curve) and the mean weight increased to 0.94 (cyan curve), as the searched speed separation increased. In contrast, as the searched speed separation increased, the slower readout speed approached the lower boundary speed (1.25°/s) (thick green curve) with the weight diminishing to negligible 0.06 (thin green curve) (*Appendix 1—figure 4F*), likely a border artifact.

Furthermore, we compared the population neural responses averaged across the one-readout-speed trials and the two-readout-speed trials. The spline-fitted population responses of the two subgroups were highly correlated ($R^2$=0.99) and statistically indistinguishable (paired t-test, *p*=0.30) (*Appendix 1—figure 4D*). This indicates that a tiny change in the population response (e.g. a slightly higher peak near log preferred speed of 3.7) would lead the decoder to exact one speed rather than two speeds (*Appendix 1—figure 4D*). In other words, the decoder was uncertain about how many speeds were in the visual stimuli and, therefore, had difficulty segmenting the visual stimuli at these fast stimulus speeds of 20 and 80°/s.

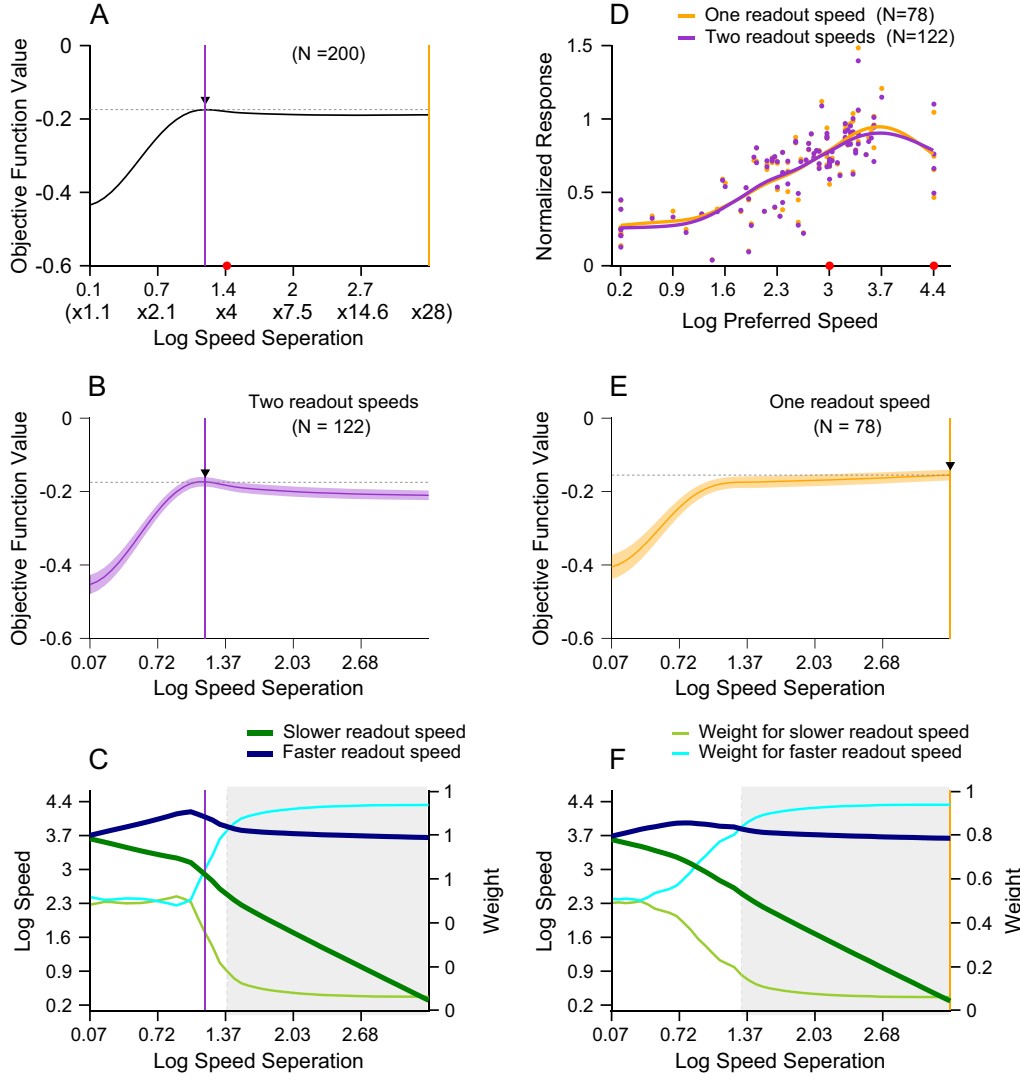

**Appendix 1—figure 4.** Analysis of decoding the bi-speed stimulus with the speeds of 20 and 80°/s. (**A**) Values of the objective function averaged across all 200 trials as the decoder searched through different speed separations. The red dot on the X-axis indicates the ground-truth speed separation of the stimulus speeds. (**B, E**) Values of the objective functions averaged across trials considered to have two (**B**) and one (**E**) readout speed(s). In A, B, and E, the error bands indicate ± STE. The black arrow indicates the speed separation where the objective function reaches its peak. The horizontal dotted line indicates the peak value of the objective function. (**C, F**) Values of the readout speeds (darker and thick lines in green and blue) and their weights (lighter and thin lines in green and cyan) as the decoder searched through different speed separations in trials considered to have two (**C**) and one (**F**) readout speed(s). (**D**) Population neural responses averaged across trials that are considered to have two readout speeds (purple) and one readout speed (orange). Each dot represents the trial-averaged response of one neuron. The curves represent the spline-fitted population neural responses. The two red dots on the X-axis indicate stimulus speeds of 20 and 80°/s.

