## [Editor Report · eLife Assessment]

This study concerns how macaque visual cortical area MT represents stimuli composed of more than one speed of motion. The study is **valuable** because little is known about how the visual pathway segments and preserves information about multiple stimuli, and the study involves perceptual reports from both humans and one monkey regarding whether there are one or two speeds in the stimulus. The study presents **compelling** evidence that (on average) MT neurons shift from faster-speed-takes-all at low speeds to representing the average of the two speeds at higher speeds. Ultimately, this study raises intriguing questions about how exactly the response patterns in visual cortical area MT might preserve information about each speed, since such information could potentially be lost in an average response as described here, depending on assumptions about how MT activity is evaluated by other visual areas.

---

## [Referee Report · Reviewer #1 (Public review)]

Summary:

Most studies in sensory neuroscience investigate how individual sensory stimuli are represented in the brain (e.g., the motion or color of a single object). This study starts tackling the more difficult question of how the brain represents multiple stimuli simultaneously and how these representations help to segregate objects from cluttered scenes with overlapping objects.

Strengths:

The authors first document the ability of humans to segregate two motion patterns based on differences in speed. Then they show that a monkey's performance is largely similar; thus establishing the monkey as a good model to study the underlying neural representations.

Careful quantification of the neural responses in the middle temporal area during the simultaneous presentation of fast and slow speeds leads to the surprising finding that, at low average speeds, many neurons respond as if the slowest speed is not present, while they show averaged responses at high speeds. This unexpected complexity of the integration of multiple stimuli is key to the model developed in this paper.

One experiment in which attention is drawn away from the receptive field supports the claim that this is not due to the involuntary capture of attention by fast speeds.

A classifier using the neuronal response and trained to distinguish single speed from bi-speed stimuli shows a similar overall performance and dependence on the mean speed as the monkey. This supports the claim that these neurons may indeed underlie the animal's decision process.

The authors expand the well-established divisive normalization model to capture the responses to bi-speed stimuli. The incremental modeling (eq 9 and 10) clarifies which aspects of the tuning curves are captured by the parameters.

---

## [Referee Report · Reviewer #3 (Public review)]

Summary:

This study concerns how macaque visual cortical area MT represents stimuli composed of more than one speed of motion.

Strengths:

The study is valuable because little is known about how the visual pathway segments and preserves information about multiple stimuli. The study presents compelling evidence that (on average) MT neurons shift from faster-speed-takes-all at low speeds to representing the average of the two speeds at higher speeds. An additional strength of the study is the inclusion of perceptual reports from both humans and one monkey participant performing a task in which they judged whether the stimuli involved one vs two different speeds. Ultimately, this study raises intriguing questions about how exactly the response patterns in visual cortical area MT might preserve information about each speed, since such information is potentially lost in an average response as described here.

Reviewing Editor comment on revised version:

The remaining concern was resolved.

---

## [Author Response]

The following is the authors’ response to the previous reviews

**Reviewer #3 (Recommendations for the authors):**
The authors have done an excellent job of addressing most comments, but my concerns about Figure 5 remain. I appreciate the authors' efforts to address the problem involving Rs being part of the computation on both the x and y axes of Figure 5, but addressing this via simulation addresses statistical significance but overlooks effect size. I think the authors may have misunderstood my original suggestion, so I will attempt to explain it better here. Since "Rs" is an average across all trials, the trials could be subdivided in two halves to compute two separate averages - for example, an average of the even numbered trials and an average of the odd numbered trials. Then you would use the "Rs" from the even numbered trials for one axis and the "Rs" from the odd numbered trials for the other. You would then plot R-Rs_even vs Rf-Rs_odd. This would remove the confound from this figure, and allow the text/interpretation to be largely unchanged (assuming the results continue to look as they do).

We have added a description and the result of the new analysis (line #321 to #332), and a supplementary figure (Suppl. Fig. 1) (line #1464 to #1477).

“We calculated 𝑅_𝑠_ in the ordinate and abscissa of Figure 5A-E using responses averaged across different subsets of trials, such that 𝑅_𝑠_ was no longer a common term in the ordinate and abscissa. For each neuron, we determined 𝑅_𝑠1_ by averaging the firing rates of 𝑅_𝑠_ across half of the recorded trials, selected randomly. We also determined 𝑅_𝑠2_ by averaging the firing rates of 𝑅_𝑠_ across the rest of the trials. We regressed (𝑅 − 𝑅_𝑠1_) on (𝑅_𝑓_ − 𝑅_𝑠2_) , as well as (𝑅_𝑠_ - 𝑅_𝑠2_) on (𝑅_𝑓_ − 𝑅_𝑠1_), and repeated the procedure 50 times. The averaged slopes obtained with 𝑅_𝑠_ from the split trials showed the same pattern as those using 𝑅_𝑠_ from all trials (Table 1 and Supplementary Fig. 1), although the coefficient of determination was slightly reduced (Table 1). For ×4 speed separation, the slopes were nearly identical to those shown in Figure 5F1. For ×2 speed separation, the slopes were slightly smaller than those in Figure 5F2, but followed the same pattern (Supplementary Fig. 1). Together, these analysis results confirmed the faster-speed bias at the slow stimulus speeds, and the change of the response weights as stimulus speeds increased.”

An additional remaining item concerns the terminology weighted sum, in the context of the constraint that wf and ws must sum to one. My opinion is that it is non-standard to use weighted sum when the computation is a weighted average, but as long as the authors make their meaning clear, the reader will be able to follow. I suggest adding some phrasing to explain to the reader the shift in interpretation from the more general weighted sum to the more constrained weighted average. Specifically, "weighted sum" first appears on line 268, and then the additional constraint of ws + wf = 1 is introduced on line 278. Somewhere around line 278, it would be useful to include a sentence stating that this constraint means the weighted sum is constrained to be a weighted average.

Thanks for the suggestion. We have modified the text as follows. Since we made other modifications in the text, the line numbers are slightly different from the last version.

Line #274 to 275:

“Since it is not possible to solve for both variables, 𝑤_𝑠_ and 𝑤_𝑓_, from a single equation (Eq. 5) with three data points, we introduced an additional constraint: 𝑤_𝑠_ + 𝑤_𝑓_ = 1. With this constraint, the weighted sum becomes a weighted average.”

Also on line #309:

“First, at each speed pair and for each of the 100 neurons in the data sample shown in Figure 5, we simulated the response to the bi-speed stimuli (𝑅_𝑒_) as a randomly weighted average of 𝑅_𝑓_ and 𝑅_𝑠_ of the same neuron.\begin{document}$$\displaystyle R_{e}=a R_{f}+(1-a) R_{s}$$\end{document}

in which 𝑎 was a randomly generated weight (between 0 and 1) for 𝑅_𝑓_, and the weights for 𝑅_𝑓_ and 𝑅_𝑠_ summed to one.”